# Thermokarst amplifies fluvial inorganic carbon cycling and export across watershed scales on the Peel Plateau, Canada

Scott Zolkos[1,*], Suzanne E. Tank[1], Robert G. Striegl[2], Steven V. Kokelj[3], Justin Kokoszka[3], Cristian Estop-Aragonés[4,†], David Olefeldt[4]

[1]Department of Biological Sciences, University of Alberta, Edmonton, AB, T6G 2E3 Canada
[2]United States Geological Survey, Boulder, CO, 80303 USA
[3]Northwest Territories Geological Survey, Yellowknife, NT, X1A 2L9 Canada
[4]Department of Renewable Resources, University of Alberta, Edmonton, AB, T6G 2E3 Canada
*Present address: Woodwell Climate Research Center, Falmouth, MA 02540 USA
[†]Present address: Institute of Landscape Ecology, University of Münster, Münster, 48149 Germany

*Correspondence to*: Scott Zolkos (sgzolkos@gmail.com)

**Abstract.** As climate warming and precipitation increase at high latitudes, permafrost terrains across the circumpolar north are poised for intensified geomorphic activity and sediment mobilization that are expected to persist for millennia. In previously glaciated permafrost terrain, ice-rich deposits are associated with large stores of reactive mineral substrate. Over geological timescales, chemical weathering moderates atmospheric $CO_2$ levels, raising the prospect that mass wasting driven by terrain consolidation following thaw (thermokarst) may enhance weathering of permafrost sediments and thus climate feedbacks. The nature of these feedbacks depend upon the mineral composition of sediments (weathering sources) and the balance between atmospheric exchange of $CO_2$ versus fluvial export of carbonate alkalinity ($\Sigma[HCO_3^-, CO_3^{2-}]$). Working in the fluvially-incised, ice-rich glacial deposits of the Peel Plateau in northwestern Canada, we determine the effects of slope thermokarst in the form of retrogressive thaw slump (RTS) activity on mineral weathering sources, $CO_2$ dynamics, and carbonate alkalinity export, and how these effects integrate across watershed scales (~2 to 1000 $km^2$). We worked along three transects in nested watersheds with varying connectivity to RTS activity: a 550 m transect along a first-order thaw stream within a large RTS; a 14 km transect along a stream which directly received inputs from several RTSs; and a 70 km transect along a larger stream with headwaters that lay outside of RTS influence. In undisturbed headwaters, stream chemistry reflected $CO_2$ from soil respiration processes and atmospheric exchange. Within the RTS, rapid sulfuric acid carbonate weathering, prompted by the exposure of sulfide- and carbonate-bearing tills, appeared to increase fluvial $CO_2$ efflux to the atmosphere and propagate carbonate alkalinity across watershed scales. Despite covering less than 1% of the landscape, RTS activity drove carbonate alkalinity to increase by two orders of magnitude along the largest transect. Amplified export of carbonate alkalinity together with isotopic signals of shifting DIC and $CO_2$ sources along the downstream transects highlight the dynamic nature of carbon cycling that may typify glaciated permafrost watersheds subject to intensification of hillslope thermokarst. The balance between $CO_2$ drawdown in regions where carbonic acid weathering predominates and $CO_2$ release in regions where sulfides are more prevalent will determine the biogeochemical legacy of thermokarst and enhanced weathering in northern permafrost terrains. Effects of RTSs on carbon cycling can be expected to persist for millennia, indicating a need for their integration into predictions of weathering-carbon-climate feedbacks among thermokarst terrains.

## 1 Introduction

Riverine export of carbonate alkalinity ($\Sigma[HCO_3^-, CO_3^{2-}]$), generated by the chemical weathering of silicate and carbonate minerals, is a key component of the global carbon cycle and Earth's long-term climate (Berner, 1999; Gaillardet et al., 1999; Hilton and West, 2020; Torres et al., 2017). The degree to which carbonate alkalinity production involves $CO_2$ (as carbonic acid, $H_2CO_3 = H_2O + CO_{2(g,aq)}$), from atmospheric or soil-respiration sources, and liberates mineral carbon influences whether dissolved inorganic carbon (DIC = $\Sigma[CO_2$, carbonate alkalinity]) in fluvial networks represents a carbon sink or source. Rapid warming at northern latitudes (Serreze and Barry, 2011) is thawing permafrost (Biskaborn et al., 2019), increasing vegetation productivity (Bjorkman et al., 2018), intensifying hydrologic cycles (Rawlins et al., 2010), and strengthening land-freshwater linkages (Vonk et al., 2019; Walvoord and Kurylyk, 2016). These processes are activating large amounts of mineral substrate into biogeochemical cycles, with significant implications for DIC cycling (Lacelle et al., 2019; Wadham et al., 2019). In recent decades, increasing riverine fluxes of carbonate alkalinity and solutes across the circumpolar north reflect enhanced mineral weathering associated with active layer thickening, deepening hydrologic flowpaths into mineral soils, and greater soil acidity from increasing vegetation productivity (Drake et al., 2018a; Tank et al., 2016; Toohey et al., 2016). Glaciated permafrost terrain hosting ice-rich deposits of reactive sediments are thought to be distributed across the northern permafrost zone, raising the prospect that terrain consolidation following thaw (thermokarst) and associated carbonate alkalinity production and export may have stronger influence on climate feedbacks in such regions (Zolkos et al., 2018).

Three coupled factors primarily influence the degree to which carbonate alkalinity represents a carbon sink or source. First, the weathering source, which accounts for both the mineral composition of substrate subjected to chemical weathering and the acid(s) responsible for weathering. Silicate weathering by $H_2CO_3$ generates alkalinity without liberating mineral carbon and thus represents a long-term $CO_2$ sink. In contrast, $H_2CO_3$ carbonate weathering is a $CO_2$ sink only over ~$10^2$–$10^3$ y, as half of the alkalinity produced is geogenic. $HCO_3^-$ produced during carbonate weathering in the presence of strong acids, for instance sulfuric acid ($H_2SO_4$) from sulfide oxidation, is a $CO_2$ source over longer timescales (~$10^6$ y; Calmels et al., 2007) and can also produce $CO_2$ over shorter timescales when $H_2SO_4$ is present in excess (Stumm and Morgan, 1996). Second, the rate of mineral weathering and processes that further alter this rate. Rates of chemical weathering are orders of magnitude faster for carbonates and sulfides than for silicates (Stumm and Morgan, 1996). Further, weathering rates generally increase with mineral surface area, and therefore are often fast in glacial environments owing to intense physical weathering (Anderson, 2007). Indeed, hydrochemical signatures of trace carbonate and sulfide lithologies can dominate weathering fluxes in primarily silicate glacial environments (Anderson, 2007). The disparity is so significant that, when sediment supplies are sufficient, $H_2CO_3$ carbonate weathering in proglacial streams can consume dissolved $CO_2$ to below atmospheric levels (Sharp et al., 1995; St. Pierre et al., 2019). Third, the magnitude of carbonate alkalinity export, which is influenced by its production via weathering of minerals during fluvial transport (e.g. Striegl et al., 2007) and its loss via carbonate equilibrium reactions and $CO_2$ degassing along the land-freshwater-ocean continuum. From a climate perspective, the magnitude of carbonate alkalinity export is particularly relevant

over geological timescales, because half of riverine carbonate alkalinity exported to the ocean is returned to the atmosphere as $CO_2$ via precipitation reactions within the marine carbon cycle (Calmels et al., 2007). Together, these three controls on carbonate alkalinity highlight the non-conservative nature of DIC and its susceptibility to transformation within fluvial networks. Hence, to constrain carbonate alkalinity export in rapidly changing permafrost terrains, nested-watershed sampling designs are critical for capturing DIC transformation along the land-freshwater-ocean continuum and resolving drivers and sources of carbon cycling across scales (Drake et al., 2018b).

Glaciated permafrost terrains are poised for rapid geomorphic and associated biogeochemical change as the climate warms and precipitation intensifies (Kokelj et al., 2017b). Despite glacial retreat across much of the circumpolar north, permafrost within these landscapes preserves biogeochemical legacies of glaciation across northern Canada, Alaska, and western Siberia (Kokelj et al., 2017b). In North America, the comminution of carbonate and shale bedrock during expansion of the Laurentide Ice Sheet (LIS) and the climate and vegetative protection of ice- and sediment-rich tills in the wake of its retreat endowed former glacial margins across northwestern Canada with thick inorganic tills held in ice-rich permafrost (Kokelj et al., 2017b). Today, the climate-driven renewal of post-glacial landscape change is mobilizing immense stores of minerals into modern biogeochemical cycles via hillslope thermokarst features, the largest of which include retrogressive thaw slumps (RTSs) (Kokelj et al., 2017a). On the Peel Plateau (NWT, Canada), RTSs expose carbonate- and sulfide-bearing glacigenic permafrost sediments that are tens of meters thick. The chemical weathering and fluvial transport of these sediments results in increased $HCO_3^-$ immediately downstream of RTSs and greater solute and sediment loads throughout downstream systems (Kokelj et al., 2013; Malone et al., 2013; Zolkos et al., 2018). RTS activity has been suggested, but not previously proved, to be partly responsible for increasing carbonate alkalinity fluxes in the larger Peel River during recent decades (Zolkos et al., 2018). Yet, it remains unknown how hillslope thermokarst effects on mineral weathering and DIC sources and cycling integrate across watershed scales on the Peel Plateau and in relatively inorganic-rich permafrost terrains across the circumpolar north. In this study we evaluated trends in major ions, DIC concentration, and dual $\delta^{13}$C-DIC–$\delta^{13}$C-$CO_2$ isotopes along transects within three nested watersheds in the Stony Creek watershed on the Peel Plateau. Our nested watershed approach was intended to allow us to determine how RTS effects on carbon cycling integrate across scales from ~1 to 1000 km$^2$.

**2 Methods**

**2.1 Study Area**

The Stony Creek watershed is located southwest of the hamlet of Fort McPherson, in the northern, or lower Peel River watershed (Fig. 1). Stony Creek, a tributary of the Peel River, originates in the Richardson Mountains, where slopes are sparsely vegetated and mainly consist of bedrock colluvium (Duk-Rodkin and Hughes, 1992). Exposed marine shale and sandstone bedrock contain sulfide- and gypsum-bearing lithologies, but limited carbonate (Norris, 1985). As Stony Creek flows eastward, the main channel and its tributaries incise ice-rich glacial deposits and underlying Cretaceous bedrock, forming a stream network comprised of tundra flow tracks grading to incised gravel bed streams with increasing watershed size. The fluvially-incised valleys and increasing regional precipitation have

proven conducive to thaw-driven mass wasting of ice-rich glacial deposits and formation of RTSs (Kokelj et al., 2017b). Growth of RTSs is driven by the ablation of exposed ground ice and is perpetuated by the downslope mass wasting of thawed material via fluidized earth flows, which can accumulate large volumes of debris in stream valleys (Fig. 1). Across the Stony Creek watershed, intensifying RTS activity releases large volumes of sediment and solutes into streams relative to undisturbed headwaters (Kokelj et al., 2017b; Segal et al., 2016). This substrate is transported to streams via rill runoff channels in the scar zone and debris tongue deposits in the stream valley. Impacts to Stony Creek are representative of numerous other major Peel River tributaries that have incised the ice-rich Peel Plateau (Kokelj et al., 2015). The ~60 km$^2$ watershed of Dempster Creek, a tributary of Stony Creek, originates in willow and open spruce taiga without RTS activity, receiving large inputs of sediments and solutes from RTSs FM2 and FM3 within several kilometers of the headwaters (Kokelj et al., 2013; Malone et al., 2013). Many small, non-RTS affected streams and several larger RTS-affected tributaries flow into Dempster Creek before its confluence with Stony Creek.

**2.2 Stream Sampling**

In late July 2017, we sampled along transects within three nested watersheds (Fig. 1, Table A1) to understand how the effects of RTSs integrate across watershed scales. (i) The *RTS FM2 runoff* transect included five sampling locations along a 550 m-long thaw stream formed by a runoff channel within an active RTS. The RTS FM2 runoff received no observable hydrologic inputs during the sampling period. (ii) A 14 km transect along the mainstem of *Dempster Creek*, which received inputs directly from RTS FM2, was sampled at one location in undisturbed headwaters and at three sites downstream of RTS FM2. Sites downstream were located on the mainstem, immediately upstream of three major tributaries. We also sampled from the tributaries near their confluence with Dempster Creek, to characterize tributary chemistry. (iii) A 70 km transect along the mainstem of *Stony Creek*, a 6[th]-order stream, was sampled at eight locations: one in undisturbed headwaters and seven on the RTS-affected reach upstream of major tributaries. We additionally sampled from one tributary of the undisturbed headwaters and from six RTS-affected tributaries near their confluence with the mainstem. Stony Creek, a major tributary of the 70000 km$^2$ Peel River watershed (Fig. 1), was sampled to determine how the effects of RTS activity on DIC integrate across broader scales.

At all sampling sites, stream temperature, specific conductance (henceforth, "conductivity"), and pH were measured using a pre-calibrated YSI Professional-Plus water quality meter. At most sites, water samples were collected for ions, DIC, $CO_2$, $CH_4$, dissolved organic carbon (DOC), UV-visible absorbance, and total suspended solids (TSS). Along the RTS FM2 runoff transect, we sampled only for DIC and $CO_2$ concentration, and stable isotopes of dissolved $CO_2$ ($\delta^{13}$C-$CO_2$). One day prior, additional parameters were sampled at RTS FM2 runoff site five, located near the confluence of the RTS runoff with Dempster Creek, for comparison with the full suite of chemistry parameters collected along the Dempster Creek transect. At the Dempster and Stony Creek sites, we additionally sampled water for stable isotopes of DIC ($\delta^{13}$C-DIC) and used dual $\delta^{13}$C-DIC–$\delta^{13}$C-$CO_2$ isotopes to characterize abiotic and biotic processes influencing DIC sources and cycling across watershed scales.

Water samples were collected from the thalweg where possible, as an integrated sample from ~15 cm below the surface to ~1 m depth. An additional sample for TSS was collected in a 1 L HDPE in the same fashion. Water samples were filtered using sample-rinsed 0.45 µm polyethersulfone (PES, ThermoFisher) or cellulose-acetate (CA, Sartorius) membranes. Samples for DIC were collected without headspace in airtight syringes. Samples for ions, DOC, and UV-visible absorbance were collected in acid washed (24 h, 10% v/v HCl) all-plastic syringes. Syringes were triple sample-rinsed, sealed without headspace, and stored cool and dark until processing within 10 h. Water for DIC was filtered (PES) into precombusted (5 h, 500°C) glass vials without headspace and sealed with a butyl rubber septum for DIC concentration or two silicone-teflon septa for $\delta^{13}$C-DIC. Samples for cations were filtered (CA) into acid-washed bottles and acidified with trace metal-grade $HNO_3$, while anions were filtered (CA) but not acidified. Samples for DOC were filtered (PES) into precombusted glass vials and acidified to pH < 2 using trace metal-grade HCl (Vonk et al., 2015). Samples for UV-visible absorbance were filtered (PES) into non-acid washed 30 mL HDPE bottles. Samples were refrigerated (4°C, dark) until analysis.

Dissolved gases were collected following the headspace equilibration method (Hesslein et al., 1991) and stored in airtight syringes (for $CO_2$ concentration) or over-pressurized in pre-evacuated serum bottles sealed with pre-baked (60°C, 12 h), gas-inert butyl rubber stoppers (for $\delta^{13}$C-$CO_2$, $CH_4$). At each site, atmospheric samples for $CO_2$ and $CH_4$ concentration and $\delta^{13}$C-$CO_2$ were stored in the same fashion. Gas samples were stored in the dark at ~20°C prior to analysis within 10 h ($CO_2$) or two months ($\delta^{13}$C-$CO_2$, $CH_4$). Water and air temperature, atmospheric pressure, and the volumetric ratio of sample to atmospheric headspace was recorded for correcting later calculations of $CO_2$ partial pressure ($p$$CO_2$) and $\delta^{13}$C-$CO_2$ (Hamilton and Ostrom, 2007).

**2.3 Hydrochemical Analyses**

Upon returning from the field each day, $CO_2$ was measured using an infrared gas analyzer (PP Systems EGM-4) which was checked monthly for drift using a commercial standard (Scotty Gases). We calculated $p$$CO_2$ using Henry's constants corrected for stream water temperature (Weiss, 1974) and accounting for the ratio of water volume to headspace during sample equilibration. DIC samples were measured by infrared $CO_2$ detection (LiCOR 7000) following acidification within a DIC analyzer (Apollo SciTech model AS-C3). Calibration curves were made daily using certified reference material (CRM) from Scripps Institution of Oceanography. Samples with DIC concentrations < 400 µM were analyzed using solutions prepared from a 1000 ppm TIC stock (ACCUSPEC) that were calibrated with CRM. DIC species ($CO_2$, $HCO_3^-$, $CO_3^{2-}$) were calculated from DIC concentration and $p$$CO_2$ or pH using CO2SYS (v.2.3) (Pierrot et al., 2006), using field temperature and pressure at the time of sampling, and the freshwater equilibrium constants for K1 and K2 (Millero, 1979).

Cations and trace elements were measured by optical emission spectrometry (Thermo ICAP-6300) and anions by ion chromatography (Dionex DX-600) at the University of Alberta Biogeochemical Analytical Services Laboratory (BASL, ISO/EIC accreditation #17025). DOC was measured using a total organic carbon analyzer (Shimadzu TOC-V). DOC standard curves were made daily with a 1000 ppm KHP solution (ACCUSPEC) and an in-house caffeine

standard (10 mg L$^{-1}$) was run every 20 samples to monitor instrument drift. Absorbance spectra were analyzed using an Ocean Optics UV-VIS instrument with a Flame spectrometer module, following Stubbins et al. (2017) and corrected for Fe interference (Poulin et al., 2014). To evaluate organic carbon reactivity, we used specific ultraviolet
absorbance at 254 nm (SUVA$_{254}$) to infer DOC aromaticity (Weishaar et al., 2003).

$\delta^{13}$C-DIC was determined using an isotope ratio mass spectrometer (Finnigan Mat DeltaPlusXP) interfaced to a total organic carbon analyzer (OI Analytical Aurora Model 1030W) at the University of Ottawa Stable Isotope Laboratory. $\delta^{13}$C-CO$_2$ and CH$_4$ concentration were analyzed within two months using a Picarro isotope analyzer (G2201-$i$; < 0.2‰ precision, CH$_4$ operational range = 1.8–1500 ppm) equipped with an injection module for discrete
samples (SSIM). Commercial $\delta^{13}$C-CO$_2$ and CH$_4$ standards were used to check for drift during each run. We used mass balance to correct $\delta^{13}$C-CO$_2$ values for the $\delta^{13}$C and mass of atmospheric CO$_2$ used for equilibration (Hamilton and Ostrom, 2007). To assess $\delta^{13}$C-CO$_2$ fractionation from calcite precipitation (Turner, 1982) and methanogenesis (Campeau et al., 2018) in RTS FM2 runoff, we calculated the saturation index (SI) and partial pressure of CH$_4$ ($p$CH$_4$). SI was calculated using the hydrochemical software Aqion version 6.7.0 (http://www.aqion.de), which uses
the U.S. Geological Survey software PHREEQC (Parkhurst and Appelo, 2013) as the internal numerical solver. Samples for atmospheric and dissolved CH$_4$ were collected in the same fashion as $\delta^{13}$C-CO$_2$. $p$CH$_4$ was calculated using Bunsen solubility coefficients (Wiesenburg and Guinasso, 1979) converted to the appropriate units (Sander, 2015).

TSS samples were filtered onto muffled (450°C, 4 h) and pre-weighed glass fiber filters (Whatman GF/F; 0.7 μm
nominal pore size) upon returning from the field, stored frozen, and dried (60°C, 24 h) for gravimetric analysis following a modified version of U.S. Geological Survey Method I-3765.

### 2.4 Mineral Weathering and DIC Sources

We used a Piper diagram (Piper, 1944), which reflects the proportional equivalent concentrations of ions in a sample relative to mineral weathering end-members, as one method to constrain the sources of mineral weathering and
HCO$_3^-$. The products of Eq. 1–7 defined the mineral weathering end-members in the Piper diagram (Table 1). We further constrained mineral weathering and DIC sources using $\delta^{13}$C-DIC and pH. End-member $\delta^{13}$C-DIC ranges for equilibrium processes (mixing with atmospheric and/or biotic CO$_2$) and kinetic reactions (mineral weathering) were derived following Lehn et al. (2017) and using published isotopic fractionation factors (Zhang et al., 1995).

To evaluate potential effects on $\delta^{13}$C-CO$_2$ from DIC speciation along the pH continuum (Eq. 8, Table 1) (Clark and
Fritz, 1997), we compared theoretical and observed $\delta^{13}$C-CO$_2$ values in the Stony Creek mainstem. Theoretical $\delta^{13}$C-CO$_2$ values were calculated using mass balance to obtain $\delta^{13}$C-HCO$_3^-$ from measurements of DIC, CO$_2$, HCO$_3^-$, $\delta^{13}$C-DIC, and $\delta^{13}$C-CO$_2$. We then used measurements of stream temperature ($T$) to calculate the equilibrium fractionation between CO$_2$ and HCO$_3^-$ ($\epsilon = -9.483 \times 10^3/T + 23.89$‰; Mook et al., 1974). Finally, $\epsilon$ was subtracted from $\delta^{13}$C-HCO$_3^-$ to obtain theoretical $\delta^{13}$C-CO$_2$. Similarity between observed and theoretical $\delta^{13}$C-CO$_2$ values was

interpreted as $\delta^{13}C$-$CO_2$ variability driven by carbonate equilibrium reactions, whereas dissimilarity was taken to

reflect effects from $CO_2$ degassing (Zhang et al., 1995) and/or biotic $CO_2$ production (Kendall et al., 2014).

**2.5 Geospatial Analyses**

Stream networks and watershed areas were delineated using the ArcHydro tools in ArcGIS 10.5 from the gridded

(30 m) Canadian Digital Elevation Model (CDEM). CDEM data were reconditioned using National Hydro Network

stream vectors, which were first modified as needed to align with stream flow paths visible in Copernicus Sentinel-2

multispectral satellite imagery (2017; European Space Agency, https://sentinel.esa.int/). To statistically assess

landscape controls on DIC cycling (Sect. 2.7), we delineated active RTSs and derived terrain roughness and

vegetation productivity in the major tributary watersheds of Stony Creek. RTSs were interpreted as active where

exposed sediment visibly dominated the feature surface (Cray and Pollard, 2015) in orthorectified SPOT

multispectral imagery that we pan-sharpened to 1.6 m resolution using the ArcGIS Image Analysis tool. The satellite

imagery was collected from September 9 to 25, 2016. Active RTSs that were connected to streams were manually

delineated using ArcGIS. We used RivEx 10.25 software (Hornby, 2017) to quantify the number of active RTSs

impacting streams in the Stony Creek watershed and to visualize the accumulation of RTS impacts across the fluvial

network. We defined RTS impact accumulation as the cumulative number of active RTSs impacting upstream

reaches. RTSs were interpreted to impact streams based on contact with the channel or interpreted downslope flow

based on slope direction and gradient from the CDEM (Supplementary Information). Where a single RTS affected

multiple streams, only the upstream segment was used for the accumulation.

We used the Geomorphic and Gradients Metrics Toolbox (Evans et al., 2014) to calculate terrain roughness, which

is a measure of variance across a land surface and represents topographic complexity (Riley et al., 1999). We use

terrain roughness as a proxy for potential physical erosion, which is known to enhance sulfide oxidation by exposing

shale regolith throughout the Peel River watershed (Calmels et al., 2007) and may therefore influence DIC. The

enhanced vegetation index (EVI) was used to broadly evaluate vegetation productivity (Huete et al., 2002), which is

known to influence DIC production by enhancing mineral weathering (Berner, 1999). We used the U.S. National

Aeronautics and Space Administration EVI product (Didan, 2015), which is derived from gridded (250 m) moderate

resolution imaging spectroradiometer (MODIS). The MODIS data were collected on July 28, 2017. The ArcGIS

Zonal Statistics tool was used to calculate total RTS area, mean terrain roughness, and mean EVI in Stony Creek

tributary watersheds.

**2.6 Stream Flow**

Water discharge ($Q$) in Stony Creek tributaries was estimated from a hydraulic geometry model (Gordon et al.,

2004) that we developed using flow measurements made in Peel Plateau streams during 2015–2017 and width ($W$)

estimated from on-site measurements or photos from 2017 with a known scale. The model reflected measurements

spanning diverse stream morphologies ($W = 0.4$–$6.6$ m) and flow conditions ($Q = 0.005$–$0.91$ $m^3$ $s^{-1}$) (Fig. A1):

$$Q = e^{\ln(W/6.258)/0.661} \ (p < 0.001, R^2 = 0.89, F_{1,18} = 150) \quad (1)$$

Discharge values from 2015 to 2017 were calculated from measurements of stream flow (RedBack Model RB1, PVD100) and cross-sectional area made at increments equal to 10% of stream width (Gordon et al., 2004; Lurry and Kolbe, 2000), and were averaged for sites with multiple measurements.

**2.7 Statistics**

We tested for downstream change in $HCO_3^-$ concentration and $pCO_2$ along the Stony Creek mainstem using the non-parametric Mann-Kendall test from the R software (R Core Team, 2018) package *zyp* (Bronaugh and Werner, 2013), following the trend pre-whitening approach detailed by Yue et al. (2002) to account for serial autocorrelation. We developed a multiple linear regression model to evaluate the influence of RTS activity on $HCO_3^-$ export in Stony Creek tributary watersheds relative to other landscape variables known to influence DIC production, including hydrology, terrain roughness, and vegetation productivity (Berner, 1992; Drake et al., 2018a). To account for potential effects of varying tributary watershed areas on discharge ($Q$) and constituent concentration, we used tributary $HCO_3^-$ yields in the model. Instantaneous discharge ($Q$, $m^3$ $s^{-1}$) was estimated from the hydraulic geometry relationship between $Q$ and stream width (Eq. 1). Discharge and $HCO_3^-$ flux (concentration*$Q$) were normalized to the respective tributary watershed area and scaled to estimate daily water yield (cm $d^{-1}$) and $HCO_3^-$ yield ($\mu$mol $m^{-2}$ $d^{-1}$). Daily $HCO_3^-$ yields in Stony Creek tributaries were modeled as:

$$HCO_3^- \ yield = RTS_n + RTS_{area} + Water \ yield + TR + EVI \quad (2)$$

where $RTS_n$ is the number of active RTSs; $RTS_{area}$ is the watershed area disturbed by RTSs (%); TR is the mean terrain roughness (m); and EVI is the mean enhanced vegetation index ($-1$ to 1). The multiple linear regression was trimmed using the *step* function in the R package *lmerTest* (Kuznetsova et al., 2018) to eliminate covariates which did not improve the model. Highly collinear covariates were identified using a Variance Inflation Factor > 3 (Zuur et al., 2010) and removed from the trimmed models. Model fits were inspected visually with residual plots and covariates were transformed as needed to meet assumptions of independent and homoscedastic residuals (Zuur, 2009). To understand potential effects from variable rainfall on water yields prior to and during the two-day sampling window of the Stony Creek tributaries, we inspected total rainfall in 24 h increments preceding the sampling of each Stony Creek tributary. Hourly rainfall data were obtained from a Government of Northwest Territories Total meteorological station located ~1 km from the RTS FM2 (Fig. A2). Statistics were performed in the R programming environment (v.3.4; R Core Team, 2018) and significance was interpreted at $\alpha = 0.05$. Summary statistics are reported as mean ± standard error, unless noted.

**3 Results**

**3.1 pH, Ions, and Weathering Sources Across Watershed Scales**

Geochemistry of the mainstem and tributary sites are summarized in Table 2. Among sites, pH was generally circumneutral and conductivity was higher in proximity to RTS activity. pH was highest in the RTS FM2 runoff ($7.69 \pm 0.05$, mean ± SE), intermediate in Dempster Creek ($7.07 \pm 0.42$), and lowest in Stony Creek ($6.86 \pm 0.21$). Along the RTS FM2 runoff transect, pH decreased from 7.72 to 7.51 between sites one and two, and increased thereafter to 7.80 at site five. pH in the Dempster Creek headwaters (5.82) was lower than in the RTS-affected reach ($7.48 \pm 0.1$). In Stony Creek pH, increased from 5.66 in headwaters to ~7.30 at sites 6–8.

Similar to pH, conductivity was higher in the RTS FM2 runoff ($1799 \pm 111$ µS cm$^{-1}$) than in Dempster Creek ($520 \pm 191$) and Stony Creek ($320 \pm 19$). Conductivity in RTS FM2 runoff increased from 1370 to 1990 µS cm$^{-1}$. Along Dempster Creek, conductivity increased from 52 µS cm$^{-1}$ in the undisturbed headwaters to 958 µS cm$^{-1}$ at the first site downstream of RTS FM2, and decreased downstream thereafter. In Stony Creek, conductivity decreased between the headwaters and the fourth downstream site, and was relatively constant at ~285 µS cm$^{-1}$ along the lower reach of Stony Creek (sites 5–8).

Streams were characterized by $Ca^{2+}$–$Mg^{2+}$–$SO_4^{2-}$-type waters (Fig. 2) with low concentrations of Cl$^-$ relative to $SO_4^{2-}$, reflecting a predominance of $H_2SO_4$ carbonate weathering and sulfate salt (e.g. gypsum) dissolution over other mineral weathering sources. A relatively greater proportion of $SO_4^{2-}$ than $HCO_3^-$ in the RTS FM2 runoff and along the upper reach of Stony Creek (sites 1–4) (Fig. 2a) suggests greater sulfate salt dissolution and/or that carbonate weathering at these sites buffered less $H_2SO_4$ (Eq. 7) than in Dempster Creek headwaters and its tributaries (Eq. 3). Along the Stony Creek mainstem (sites 1–8), increasing $HCO_3^-$ (Fig. 2a) reflected inputs from RTS-affected tributaries (sites 2–7) having relatively more $HCO_3^-$-type waters (Fig. 2b) from $H_2SO_4$ and potentially $H_2CO_3$ carbonate weathering.

**3.2 $HCO_3^-$ Concentration and $p$CO$_2$**

Carbonate alkalinity ($HCO_3^-$ + $CO_3^{2-}$) was primarily $HCO_3^-$ (>99%) at all sites. $HCO_3^-$ was highest in the RTS FM2 runoff ($1429 \pm 23$ µM), intermediate in Dempster Creek ($864 \pm 261$ µM), and lowest in Stony Creek ($255 \pm 59$ µM). Along the RTS FM2 runoff transect, $HCO_3^-$ decreased from 1510 to 1386 µM. In Dempster Creek and Stony Creek, $HCO_3^-$ concentrations were relatively low in undisturbed headwaters (115 and 33 µM, respectively) and two to ten times higher at the first RTS-affected site (1321 and 69 µM, respectively). $HCO_3^-$ decreased along the entire RTS-affected reach of Dempster Creek (from 1321 to 946 µM) in conjunction with inputs from dozens of tributary watersheds without active RTSs. In contrast, $HCO_3^-$ increased significantly along Stony Creek ($p < 0.01$, Mann-Kendall test) (Fig. 3a) in conjunction with inputs from RTS-affected tributaries.

$CO_2$ was oversaturated at all sites (Fig. 3b) and a minor component of DIC (<10%) at most sites except the undisturbed headwaters of Dempster Creek (site 1) and upper Stony Creek (sites 1–3). $p$CO$_2$ was highest in the Dempster Creek headwaters (2467 µatm), relatively high in the RTS FM2 runoff ($1023 \pm 137$ µatm), and consistently near atmospheric levels along Stony Creek ($479 \pm 12$ µatm). Along the RTS FM2 runoff transect, $p$CO$_2$ increased from 1046 to 1534 µatm within the first 220 m and then decreased from 1534 to 742 µatm over the final

m. Along Dempster Creek, $p$CO$_2$ decreased from 2467 in the headwaters to 686 µatm at the first RTS-affected site, and further decreased to 600 µatm by the end of Dempster Creek. $p$CO$_2$ in Dempster and Stony Creek tributaries were generally similar to the mainstem sites.

### 3.3 DOC Concentration and SUVA$_{254}$

DOC concentrations were highest in Dempster Creek (933 ± 83 µM), intermediate in the RTS FM2 runoff (758 ± 152 µM), and lowest in Stony Creek (303 ± 54 µM). Along the Dempster Creek transect, DOC decreased between the undisturbed headwaters (960 µM) and the first RTS-affected site (790 µM) and increased thereafter along the transect (to 1156 µM) (Fig. 3c). Along Stony Creek, DOC increased significantly (from 102 to 551 µM) ($p < 0.001$, Mann-Kendall test).

SUVA$_{254}$ values were lowest in the RTS FM2 runoff (1.85 ± 0.4 L mgC$^{-1}$ m$^{-1}$), highest in Dempster Creek (3.10 ± 0.2 L mgC$^{-1}$ m$^{-1}$), and intermediate in Stony Creek (2.51 ± 0.3 L mgC$^{-1}$ m$^{-1}$). SUVA$_{254}$ values along the Dempster Creek transect followed a similar pattern to DOC and along Stony Creek SUVA$_{254}$ values doubled (Fig. 3d). DOC and SUVA$_{254}$ increased in consecutive downstream tributaries of Stony Creek, but not Dempster Creek.

### 3.4 Stable Isotopic Composition of Carbon in DIC and CO$_2$

$\delta^{13}$C-DIC values were highest in the RTS FM2 runoff (–1.0‰) and lower, on average, along the mainstem Dempster Creek (–7.5 ± 2.5‰) and Stony Creek (–8.4 ± 0.5‰). In the undisturbed headwaters of Dempster and Stony Creek, relatively negative $\delta^{13}$C-DIC values (–11.6 to –15.6‰) reflected DIC sourced from a combination of atmospheric and biogenic (soil) CO$_2$ (Fig. 4). In the RTS FM2 runoff, relatively $^{13}$C-enriched $\delta^{13}$C-DIC (–1.0‰) aligned with H$_2$SO$_4$ carbonate weathering. $\delta^{13}$C-DIC decreased from –4.2‰ at the first site downstream of the RTS FM2 runoff to –5.7‰ at the end of Dempster Creek. Along Stony Creek, $\delta^{13}$C-DIC increased from the undisturbed headwaters (–11.6‰) to the most downstream site (–7.8‰). $\delta^{13}$C-DIC signals of H$_2$SO$_4$ carbonate weathering diminished slightly downstream along the Dempster Creek transect and intensified along Stony Creek (Fig. 4).

Similar to $\delta^{13}$C-DIC, $\delta^{13}$C-CO$_2$ values were higher in the RTS FM2 runoff (–11.0 ± 0.4‰) than along the mainstem Dempster Creek (–17.9 ± 1.4‰) and Stony Creek (–16.7 ± 0.6‰) (Fig. 5). $\delta^{13}$C-CO$_2$ values were relatively low in the undisturbed headwaters of Dempster Creek (–21.6‰), and intermediate in the headwaters of Stony Creek (–13.8‰) (Fig. 5). Along the RTS FM2 runoff transect, $\delta^{13}$C-CO$_2$ values increased from sites one to four (–12.1 to –10.0‰) and decreased at site five (–11.2‰). Along the RTS-affected reach of Dempster Creek, $\delta^{13}$C-CO$_2$ values decreased from –16.0 to –18.5‰ in conjunction with inputs from non RTS-affected tributaries having relatively low $\delta^{13}$C-CO$_2$ (–18.7 ± 1.4‰) that was more similar values from soil-respired CO$_2$. Along Stony Creek, $\delta^{13}$C-CO$_2$ values decreased from –13.8 to –18.1‰, showing a trend opposite that of $\delta^{13}$C-DIC (Fig. 5). Among sites, atmospheric $\delta^{13}$C-CO$_2$ values were relatively consistent (–9.5 ± 0.4‰, mean ± SD).

Variance in $\delta^{13}C$ of $CO_2$ and DIC could be influenced by biotic production, $CO_2$ conversion to $HCO_3^-$, and/or

mixing with atmospheric $CO_2$. To evaluate the relative influence of these processes, we compared measured $\delta^{13}C$-$CO_2$ for Stony Creek with theoretical values reflecting DIC controlled by speciation along the pH continuum (Sect. 2.4). In the undisturbed headwaters, $\delta^{13}C$-$CO_2$ indicated stronger influence from atmospheric $CO_2$ (Fig. 6). Along the upper, RTS-affected reach of Stony Creek (sites 2–5, from ~5 to 35 km), the good agreement between measured and theoretical $\delta^{13}C$-$CO_2$ values reflected equilibrium fractionation ($\varepsilon = 9.7‰$ at $9°C$) (Mook et al., 1974) between

$CO_2$ and $HCO_3^-$, indicating greater influence from DIC speciation (Fig. 6). Along the lower RTS-affected reach of the transect (sites 6–8), $\delta^{13}C$-$CO_2$ values more strongly reflected biotic $CO_2$ production with potential effects from degassing and/or $CO_2$ conversion to $HCO_3^-$. These trends in $\delta^{13}C$-$CO_2$ values along Stony Creek show a downstream change in the processes influencing DIC source, which may be related to inputs of weathering solutes and organic matter from RTS-affected tributaries.

**3.5 Stony Creek Tributary Carbonate Alkalinity Yields and Watershed Characteristics**

Carbonate alkalinity yields in RTS-affected tributaries of Stony Creek ($1558 \pm 1135$ $\mu$mol m$^{-2}$ d$^{-1}$, mean $\pm$ SD) were three orders of magnitude higher than in the non-RTS affected headwaters (1.8 $\mu$mol m$^{-2}$ d$^{-1}$) (Table 3). Consecutive downstream tributary watersheds exhibited no clear trends in the number of RTSs, the area disturbed by RTSs, terrain roughness, or EVI. In the Stony Creek headwater tributary, which had no active RTSs, terrain roughness

(16.2 m) and vegetation productivity (EVI = 0.28) were higher than in the other six tributary watersheds ($4.3 \pm 1.3$ m and $0.46 \pm 0.01$, mean $\pm$ SD, respectively). In the other tributary watersheds, the number of active RTSs reached 50 ($15 \pm 17$, mean $\pm$ SD) and RTS disturbance area reached 3.5% ($0.91 \pm 1.29\%$, mean $\pm$ SD) (Table 3).

To elucidate landscape controls on carbonate alkalinity export in Stony Creek tributary watersheds, we paired geospatial data for active RTSs, terrain roughness, and vegetation productivity with estimates of carbonate alkalinity

and water yields in a multiple linear regression model (Sect. 2.7). Water yield and the area of RTS disturbance were retained during automated covariate selection for the final model ($F_{2,4} = 63$, $p < 0.001$, $R^2 = 0.95$). In addition to the expected relationship between water yield and carbonate alkalinity yield, RTS disturbance area was a clear, significant predictor of carbonate alkalinity yield and formed a stronger relationship with alkalinity than did water yield (Table 3).

**4 Discussion**

**4.1 Rapid Carbon Cycling in Fluvial Network Headwaters**

Within undisturbed headwaters and RTS runoff on the Peel Plateau, rapid carbon cycling enhanced fluvial $CO_2$ efflux to the atmosphere. In undisturbed headwaters, $\delta^{13}C$-$CO_2$ values indicate inputs of primarily biogenic $CO_2$ from soil respiration into Dempster Creek. In the Stony Creek headwaters, intermediate $\delta^{13}C$-$CO_2$ and $pCO_2$

saturation suggested influence from exchange with atmospheric $CO_2$ and from some biogenic $CO_2$. In the

undisturbed Dempster Creek headwaters, a 70% decrease in $p\text{CO}_2$ within several kilometers downstream likely reflected degassing and diminishing inputs of respired $\text{CO}_2$ from soils to streams, relative to headwaters (Hutchins et al., 2019). These trends resemble headwater streams elsewhere, in that hydrologic inputs of respired $\text{CO}_2$ from riparian soils can drive $\text{CO}_2$ supersaturation in fluvial network headwaters (Campeau et al., 2018; Crawford et al., 2013), which is rapidly effluxed to the atmosphere over short distances downstream (Hotchkiss et al., 2015). In contrast, trends in hydrochemistry and stable isotopes within RTS FM2 runoff demonstrate that drivers of carbon cycling within RTSs are starkly different from those in undisturbed headwaters on the Peel Plateau.

Along the RTS FM2 runoff transect, the increase in conductivity corroborates experimental evidence (Zolkos and Tank, 2020) that permafrost sediments on the Peel Plateau can rapidly weather during fluvial transport within runoff. In the upper reach of the runoff transect, near RTS FM2, the decrease in $\text{HCO}_3^-$, increase in $\text{CO}_2$, and relatively enriched $\delta^{13}\text{C-CO}_2$ (Fig. 5) indicate rapid production of geogenic $\text{CO}_2$ via $\text{H}_2\text{SO}_4$ carbonate weathering (Eq. 7) and carbonate equilibrium reactions (Eq. 8). In Yedoma terrains in Siberia and Alaska, where mineral soils are relatively more organic-rich, thermokarst is associated with production of biogenic $\text{CO}_2$ (Drake et al., 2018b). While respiration likely produced some $\text{CO}_2$ in RTS FM2 runoff (Littlefair et al., 2017), observed $\delta^{13}\text{C-CO}_2$ (–11‰) more strongly reflected $\text{H}_2\text{SO}_4$ weathering of regional carbonate bedrock (–0.7 to –5.6‰) (Hitchon and Krouse, 1972) when accounting for isotopic fractionation of ~8‰ between carbonate and $\text{CO}_2$ at the temperature of FM2 runoff (18°C) (Clark and Fritz, 1997). Along the lower reach of the FM2 runoff transect, the increase in $\delta^{13}\text{C-CO}_2$ aligned with the preferential loss of $^{12}\text{C}$ in the $\text{CO}_2$ phase via DIC fractionation and degassing (Doctor et al., 2008; Drake et al., 2018b; Kendall et al., 2014). $^{13}\text{C}$ enrichment of the $\text{CO}_2$ pool by methanogenesis (Campeau et al., 2018), photosynthesis (Descolas-Gros and Fontungne, 1990), and/or calcite precipitation (Turner, 1982) was unlikely, as $\text{CH}_4$ in FM2 runoff was relatively low ($p\text{CH}_4 = 3.6 \pm 1.9$ µatm, mean $\pm$ SD, $n = 6$), the high turbidity of FM2 runoff likely inhibited photosynthesis (Levenstein et al., 2018), and calcite was below saturation (SI = –0.79). These trends demonstrate that weathering of sediments during fluvial transport within RTS runoff can result in rapid $\text{CO}_2$ production and efflux to the atmosphere, in agreement with recent estimates of high rates of $\text{CO}_2$ efflux within RTS runoff (Zolkos et al., 2019).

High rates of weathering within RTS FM2 runoff aligns with observations of substantial solute production via the exposure and weathering of carbonate flour in glacial foreground environments (Anderson, 2007; Sharp et al., 1995; St. Pierre et al., 2019). Because minerals exposed by deeper RTSs are generally reactive, and sediment concentrations increased by three orders of magnitude between the undisturbed Dempster Creek headwaters and the first RTS-affected site, we reasoned that $\text{H}_2\text{CO}_3$ weathering of these sediments during fluvial transport would measurably influence $p\text{CO}_2$ along Dempster Creek (Eq. 1) (St. Pierre et al., 2019; Striegl et al., 2007). Although $p\text{CO}_2$ decreased along the RTS-affected reach of the Dempster Creek transect (sites 2–4, Fig. 2b), coincident decreases in conductivity, $\text{HCO}_3^-$, and pH (Table 2, Figs. 2a, 4) suggest that degassing and dilution associated with inputs from non RTS-affected tributaries had stronger effects on $p\text{CO}_2$ than did $\text{H}_2\text{CO}_3$ carbonate weathering, even at the relatively short scale of this 14 km transect. From a carbon cycling perspective, biogeochemically reactive

mineral substrate appears to be rapidly transformed in headwaters on the Peel Plateau; geogenic $CO_2$ production is relegated to within RTSs; and more stable weathering products, including alkalinity, are exported downstream.

**4.2 RTS Activity in Headwaters Amplifies Carbonate Alkalinity Production and Accumulation Across Scales**

Similar to $CO_2$, alkalinity production on the Peel Plateau was strongly coupled to primarily $H_2SO_4$ carbonate
weathering mediated by RTS activity. This was reflected by a modest decrease in $HCO_3^-$ along Dempster Creek in tandem with decreasing RTS disturbance area (from 3.2 to 1.2%) and some dilution by inputs from non RTS-affected tributaries. Multiple linear regression results further indicated that RTS activity was a primary terrain control on carbonate alkalinity yields. In the Stony Creek headwaters, low carbonate alkalinity yield relative to water yield suggested that $HCO_3^-$ export was limited by carbonate availability rather than by water. In RTS-affected tributaries, higher carbonate alkalinity yields relative to water yields aligned with the model results indicating that RTS activity increases carbonate weathering and alkalinity export beyond what would otherwise be expected on the Peel Plateau. $HCO_3^-$ yields in RTS-affected tributaries were comparable to summertime $HCO_3^-$ yields in watersheds with carbonate rock weathering by glacial activity (~3000 $\mu mol\ m^{-2}\ d^{-1}$) (Lafrenière and Sharp, 2004; Striegl et al., 2007), emphasizing that unmodified sulfide- and carbonate-bearing sediments in regional permafrost are highly reactive (Zolkos and Tank, 2020) and primary sources of DIC within intermediate-sized (1000 $km^2$) fluvial networks. This aligns with stable sulfur isotopes in RTS runoff and near the Stony Creek outflow that strongly reflected sulfide oxidation (Zolkos et al., 2018). Unlike $CO_2$, the increase in $HCO_3^-$ by orders of magnitude along Stony Creek in association with inputs from RTS-affected tributaries shows that more chemically stable (i.e. non-gaseous) weathering products accumulated across scales. This aligns with previous findings that solutes and sediments from RTSs propagate through fluvial networks (Kokelj et al., 2013; Malone et al., 2013), and suggests that future intensification of RTS activity (Segal et al., 2016) will increase $HCO_3^-$ export to downstream environments.

**4.3 Integration of RTS Effects on Carbon Cycling Across Watershed Scales**

These findings enable us to develop a conceptual model of catchment chemical characteristics and how the effects of RTS activity on carbon cycling integrate across watershed scales on the Peel Plateau (Fig. 7). This model may be generalized to permafrost terrains elsewhere for testing hypotheses related to thermokarst effects on carbon cycling across the land-freshwater-ocean continuum (Tank et al., 2020).

In undisturbed headwaters on the Peel Plateau, DIC was primarily $CO_2$ and sources of $CO_2$ varied from relatively more atmospheric in the sparsely-vegetated and mountainous Stony Creek headwaters (Fig. 7a i), to more biogenic in the tundra-taiga headwaters of Dempster Creek (Fig. 7a ii). Downstream, $CO_2$ loss and mixing of streams resulted in undisturbed headwaters having relatively modest DIC comprised of a relatively large proportion of $CO_2$ sourced from mixing with the atmosphere and likely some inputs from soil respiration (Fig. 7a iii). Underlying the trends in $CO_2$ concentration, measurements of $\delta^{13}C\text{-}CO_2$ revealed shifting sources of $CO_2$ across scales (discussed below).

Thaw and exposure of reactive tills (Lacelle et al., 2019; Zolkos and Tank, 2020) by RTS activity in Peel Plateau
headwaters (see also Kokelj et al., 2013; Malone et al., 2013) promotes mineral weathering, rapidly generating $CO_2$
and substantial alkalinity. Alkalinity, along with large amounts of sediment (van der Sluijs et al., 2018) and organic
matter (Shakil et al., 2020), are exported from RTSs into fluvial networks (Fig. 7a iv). Similar to other locations,
DOC in RTS runoff on the Peel Plateau is known to be relatively biolabile (Littlefair et al., 2017), suggesting inputs
from RTS FM2 to larger streams (e.g. Dempster Creek) could stimulate biotic $CO_2$ production.

$CO_2$ degassing is most pronounced within RTSs and in undisturbed headwaters that are strongly coupled with soil
respiration, and active mineral weathering is less pronounced in mid-order streams (e.g. Dempster Creek). Hence,
mid-order streams, which also mix with inputs from undisturbed tributaries, export $HCO_3^-$ downstream at a
magnitude coupled to the area of RTS disturbance (Fig. 7a v). Further, immediately downstream of the RTS FM2
inflow to Dempster Creek, the decrease in $CO_2$ and shift in $\delta^{13}C$-$CO_2$ away from a biotic source suggest that $CO_2$
degassing to the atmosphere was more prominent than respiration of permafrost DOC (Doctor et al., 2008; Drake et
al., 2018b; Kendall et al., 2014). Thus, immediately downstream of RTSs, microbial respiration of permafrost DOC
does not appear to generate substantial $CO_2$. This may be due to lower rates of DOC mineralization than degassing,
and/or the protection of DOC from microbial oxidation via adsorption to RTS sediments (Gentsch et al., 2015). The
latter aligns with the observed decrease in DOC concentration and increase in TSS (to 11800 mg $L^{-1}$) between
Dempster Creek sites one and two (Table 2, Fig. 3c) (see also Littlefair et al., 2017). However, these effects may
diminish farther downstream in mid-order streams. Along the lower reach of Dempster Creek (sites 3–4), the
decrease in $\delta^{13}C$-$CO_2$, increase in DOC, and SUVA$_{254}$ resembling terrestrial-origin DOC from tributary streams
suggest that undisturbed tributary streams may deliver biogenic $CO_2$ and/or stimulate organic matter respiration in
RTS-affected streams. Thus, effects of RTS sediments on $CO_2$ are attenuated downstream as DOC inputs increase.

Up to and likely beyond scales of ~$10^3$ km$^2$ (e.g. Stony Creek), the largest scale of this study, $HCO_3^-$ concentrations
are likely to increase significantly downstream, reflecting the export of relatively stable weathering products (see
also Kokelj et al., 2013; Malone et al., 2013; Zolkos et al., 2018) and accumulation of carbonate alkalinity (Fig. 7a
vi, Fig. 7b). These effects were primarily driven by inputs of $HCO_3^-$ from RTS-affected tributaries, which also
increased DOC significantly along Stony Creek. Potentially owing to organic matter limitation, $CO_2$ in the
undisturbed headwaters of Stony Creek appeared to be driven by relatively faster carbonate equilibrium reactions
(Eq. 8) (Stumm and Morgan, 1996). In contrast, along the lower RTS-affected reach of Stony Creek, as $HCO_3^-$ and
DOC increased and pH stabilized, $\delta^{13}C$-$CO_2$ measurements suggest that respiration of organic matter from RTS-
affected tributaries contributed to $CO_2$ oversaturation (Fig. 6). In higher-orders streams within RTS-affected fluvial
networks, biotic $CO_2$ production may increase together with $HCO_3^-$ concentrations. This trend was not evident in
$\delta^{13}C$-DIC, which primarily reflected inputs of geogenic DIC from RTS-affected tributaries. Thus, sources of $CO_2$
may shift across scales in RTS-affected fluvial networks, and measurements of $\delta^{13}C$-$CO_2$ highlight a decoupling
between the drivers of $CO_2$ and $HCO_3^-$ at larger scales (Horgby et al., 2019; Hutchins et al., 2020). A stronger signal
of biogenic $CO_2$ production in larger streams than in permafrost thaw streams within thermokarst, as we observed, is
opposite common trends in Yedoma terrains (Drake et al., 2018b) and may partly reflect limitation of organic

substrate in Stony Creek headwaters that is relieved by RTS inputs farther downstream (Shakil et al., 2020). Underlying these trends, RTS disturbance area increased along the Stony Creek transect, from 0% in the undisturbed headwaters to 0.36% in the tributary watersheds (sites 4–8). Despite RTS activity occupying a small proportion of the landscape, carbonate alkalinity propagated through fluvial networks. These findings directly link intensifying RTS activity on the Peel Plateau (Segal et al., 2016) with signals of increasing weathering and carbonate alkalinity

export in the broader Peel and Mackenzie River watersheds (Tank et al., 2016; Zolkos et al., 2018).

**4.4 Implications for Carbon Cycling in Northern Permafrost Regions**

Permafrost terrains susceptible to hillslope thermokarst like RTSs occur within and outside of former glacial limits across the circumpolar north (Olefeldt et al., 2016; Zolkos et al., 2018), and variability in geology, glacial activity, climate, and ecosystem history cause permafrost mineral composition to vary between regions. The degree to which

carbonate weathering is coupled with sulfide oxidation will determine if mineral weathering is a $CO_2$ sink (Eq. 1) or source (Eqs. 3,7) over the coming millennia (Zolkos et al., 2018). Where thermokarst releases inorganic substrate with limited prior modification – as in larger RTSs on the Peel Plateau (Lacelle et al., 2019; Zolkos and Tank, 2020) – carbon cycling can be expected to be rapid and driven by inorganic processes, and strengthen abiotic components of the permafrost carbon-climate feedback (Schuur et al., 2015). Current dynamic-numerical biogeochemical models

for the Mackenzie River basin suggest the ubiquity of sulfide minerals reduces weathering consumption of atmospheric $CO_2$ by half (Beaulieu et al., 2012). These models do not account for enhanced $H_2SO_4$ carbonate weathering associated with RTS activity, which our results show is significantly and positively correlated with alkalinity production and export across watershed scales. Further, climate feedbacks associated with RTS activity appear to be scale-dependent. RTSs rapidly generate $CO_2$, but its outgassing occurs mostly within runoff and

comprises a small proportion of watershed-scale fluvial $CO_2$ efflux (Zolkos et al., 2019). Carbonate alkalinity generated within RTSs represents a much larger positive feedback to climate change, albeit over geological timescales, via carbonate precipitation reactions within the marine carbon cycle (Calmels et al., 2007). Future intensification of RTS activity (Segal et al., 2016) can thus be expected to increase geogenic $CO_2$ production within headwaters (see also Zolkos et al., 2019) and carbonate alkalinity export across scales (Fig. 7b) (Tank et al., 2016).

Cross-scale watershed investigations will help to understand these effects across terrains with varying lithologies and permafrost composition, and the implications of hillslope thermokarst for climate feedbacks.

**5 Conclusions**

Climate-driven renewal of geomorphic activity across permafrost preserved glacigenic terrain and associated carbonate weathering in the western Canadian Arctic is amplifying aquatic carbon export across scales, despite

RTSs disturbing only a fractional proportion of the landscape. Primary consequences include geogenic $CO_2$ production that is rapid, and localized to RTSs, and augments soil-respired $CO_2$ efflux from undisturbed headwater streams. Significant carbonate alkalinity production and export from RTSs project through fluvial networks and likely to Arctic coastal marine environments, forecasting stronger land-freshwater-ocean linkages (Tank et al., 2016)

as RTS activity intensifies in glacial margin landscapes across northwestern Canada (Kokelj et al., 2017a). Legacy effects of RTSs on carbon cycling can be expected to persist for millennia, indicating a need for the integration of dynamic-numerical biogeochemical models (Beaulieu et al., 2012) into predictions of weathering-carbon-climate feedbacks (Zolkos et al., 2018) among northern thermokarst terrains (Turetsky et al., 2020).

*Data availability*.  All data used in this study are available in the supplement.

*Supplement*.  The supplement for this article is available online.

*Author contributions*.  SZ and SET designed the study with contribution from RGS and SVK. SZ led the field research, laboratory analyses, and manuscript writing. JK contributed to geospatial analyses. CEA contributed to laboratory analyses. All authors (SZ, SET, RGS, SVK, JK, CEA, DO) contributed to manuscript writing.

*Competing interests*.  The authors declare no conflicts of interest.

*Acknowledgements*.  Research was supported by the Natural Sciences and Engineering Research Council of Canada, Campus Alberta Innovates Program, Natural Resources Canada Polar Continental Shelf Program, Environment Canada Science Youth Horizons, UAlberta Northern Research Award, and Arctic Institute of North America Grant-in-Aid. We thank Rosemin Nathoo, Christine Firth, Dempster Collin, Abraham Snowshoe, Sarah Shakil, and Erin MacDonald for assistance in the field. NWT Geological Survey contribution #. Any use of trade, product or firm names in this publication is for descriptive purposes only and does not imply endorsement by the U.S. Government.

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

**Table 1.** Mineral weathering equations used to create Piper diagram end-members. $H_2CO_3$ = carbonic acid, $H_2SO_4$ =
sulfuric acid. $H_2CO_3$ includes dissolved $CO_{2(g)}$.

| Eq. | Reaction | Equation | Reference |
|---|---|---|---|
| 1 | $H_2CO_3$ carbonate weathering (CACW) | $H_2CO_3 + (Ca,Mg)CO_3 \rightarrow (Ca^{2+},Mg^{2+}) + 2HCO_3^-$ | Lehn et al. (2017) |
| 2 | $H_2CO_3$ silicate weathering (CASW) | $2H_2CO_3 + 3H_2O + (Ca,Mg)Al_2Si_2O_8 \rightarrow (Ca^{2+},Mg^{2+}) + 2HCO_3^- + 2Al_2Si_2O_5(OH)_4$ | Lehn et al. (2017) |
| 3 | $H_2SO_4$ carbonate weathering (SACW) | $H_2SO_4 + 2(Ca,Mg)CO_3 \rightarrow 2(Ca^{2+},Mg^{2+}) + SO_4^{2-} + 2HCO_3^-$ | Lehn et al. (2017) |
| 4 | $H_2SO_4$ silicate weathering (SASW) | $H_2SO_4 + H_2O + (Ca,Mg)Al_2Si_2O_8 \rightarrow (Ca^{2+},Mg^{2+}) + SO_4^{2-} + Al_2Si_2O_5(OH)_4$ | Lehn et al. (2017) |
| 5 | Sulfate salt dissolution (SSD) | $(Ca,Mg)SO_4 \rightarrow (Ca^{2+},Mg^{2+}) + SO_4^{2-}$ | Lehn et al. (2017) |
| 6 | Sulfide oxidation | $FeS_2 + 15/4O_2 + 7/2H_2O \rightarrow Fe(OH_3) + 2H^+ + SO_4^{2-}$ | Calmels et al. (2007) |
| 7 | Carbonate weathering by $H_2SO_4$ in excess (SA$_{ex}$CW) | $2H_2SO_4 + CaMg(CO_3)_2 \rightarrow Ca^{2+} + Mg^{2+} + 2SO_4^{2-} + 2H_2CO_3$ | Stallard and Edmond (1983) |
| 8 | Carbonate equilibrium / DIC speciation | $H_2O + CO_{2(g)} \rightleftharpoons H_2CO_3 \rightleftharpoons H^+ + HCO_3^- \rightleftharpoons 2H^+ + CO_3^{2-}$ | Stumm and Morgan (1996) |

**Table 2.** Geochemistry of mainstem and tributary sites along Dempster and Stony Creeks. Retrogressive thaw slump
(RTS) FM2 runoff samples collected on July 31, 2017, except where noted ([†]July 30, 2017). RTS FM2 runoff site #5
was nearest the confluence with Dempster Creek (Fig. 1). Area = watershed area, SE = standard error. *Not RTS-
affected.

| Type | Site | pH | Cond ($\mu$S cm$^{-1}$) | $p$CO$_2$ ($\mu$atm) | CO$_2$ ($\mu$M) | HCO$_3^-$ ($\mu$M) | CO$_3^{2-}$ ($\mu$M) | DIC (uM) | $\delta^{13}$C-DIC (‰VPDB) | $\delta^{13}$C-CO$_2$ (‰VPDB) | DOC ($\mu$M) | SUVA$_{254}$ (L mgC$^{-1}$ m$^{-1}$) | TSS (mg L$^{-1}$) | Area (km$^2$) | RTS (% area) |
|---|---|---|---|---|---|---|---|---|---|---|---|---|---|---|---|
| RTS FM2 (Runoff) | 1 | 7.72 | 1370 | 1046 | 43 | 1510 | 5.2 | 1559 | – | -12.1 | 758 | 1.85 | – | – | – |
| | 2 | 7.51 | 1816 | 1534 | 60 | 1439 | 3.5 | 1502 | – | -11.4 | – | – | – | – | – |
| | 3 | 7.71 | 1920 | 914 | 37 | 1419 | 5.4 | 1462 | – | -10.3 | – | – | – | – | – |
| | 4 | 7.73 | 1903 | 878 | 38 | 1391 | 5.1 | 1433 | – | -10.0 | – | – | – | – | – |
| | 5 | 7.80 | 1986 | 742 | 33 | 1386 | 5.8 | 1424 | – | -11.2 | – | – | – | – | – |
| | 5[†] | 7.82 | 1653 | 691 | 29 | 1450 | 7.1 | 1487 | -1.0 | -11.6 | 726 | 1.84 | 15805 | – | – |
| | Mean | 7.69 | 1799 | 1023 | 42 | 1429 | 5.01 | 1476 | | -11.0 | 758 | 1.85 | | | |
| | (SE) | (0.05) | (111) | (137) | (5) | (23) | (0.39) | (25) | – | (0.4) | (152) | (0.37) | – | – | – |
| Dempster (Mainstem) | 1* | 5.82 | 52 | 2467 | 124 | 115 | 0.0 | 239 | -15.0 | -21.6 | 960 | 3.66 | 5 | 2 | 0.00 |
| | 2 | 7.55 | 958 | 686 | 35 | 1321 | 4.8 | 1361 | -4.2 | -16.0 | 790 | 2.53 | 11795 | 16 | 3.18 |
| | 3 | 7.54 | 655 | 656 | 31 | 1073 | 3.6 | 1107 | -5.3 | -15.6 | 823 | 2.93 | 9165 | 24 | 2.18 |
| | 4 | 7.35 | 416 | 600 | 30 | 946 | 2.9 | 978 | -5.7 | -18.5 | 1156 | 3.28 | 2797 | 57 | 1.19 |
| | Mean | 7.07 | 520 | 1102 | 55 | 864 | 2.8 | 921 | -7.5 | -17.9 | 933 | 3.10 | 5940 | 25 | 1.64 |
| | (SE) | (0.42) | (191) | (455) | (23) | (261) | (1.0) | (241) | (2.5) | (1.4) | (83) | (0.24) | (2735) | (12) | (0.68) |
| Dempster (Tributary) | 2* | 7.56 | 390 | 836 | 50 | 1233 | 2.8 | 1286 | -10.5 | -21.3 | 1053 | 3.46 | 26 | 2 | 0.00 |
| | 3 | 7.32 | 171 | 478 | 23 | 561 | 1.3 | 586 | -7.2 | -18.1 | 1241 | 3.65 | 985 | 11 | 1.47 |
| | 4 | 7.30 | 236 | 552 | 27 | 697 | 1.7 | 726 | -8.0 | -16.5 | 922 | 3.61 | 223 | 168 | 0.40 |
| | Mean | 7.39 | 266 | 622 | 34 | 830 | 1.9 | 866 | -8.6 | -18.7 | 1072 | 3.57 | 411 | 61 | 0.62 |
| | (SE) | (0.08) | (65) | (109) | (8) | (205) | (0.4) | (214) | (1.0) | (1.4) | (93) | (0.06) | (292) | (54) | (0.44) |
| Stony (Mainstem) | 1* | 5.66 | 406 | 543 | 33 | 33 | 0.0 | 65 | -11.6 | -13.8 | 102 | 1.29 | 3 | 83 | 0.00 |
| | 2 | 6.37 | 396 | 448 | 25 | 69 | 0.0 | 94 | -8.0 | -15.1 | 124 | 1.58 | 920 | 136 | 0.01 |
| | 3 | 7.01 | 334 | 473 | 27 | 112 | 0.0 | 139 | -6.9 | -15.3 | 202 | 2.16 | 799 | 176 | 0.27 |
| | 4 | 6.69 | 283 | 444 | 25 | 248 | 0.2 | 273 | -8.9 | -17.6 | 306 | 2.77 | 462 | 479 | 0.39 |
| | 5 | 7.20 | 279 | 482 | 27 | 325 | 0.4 | 353 | -8.4 | -17.6 | 364 | 3.09 | 507 | 490 | 0.38 |
| | 6 | 7.33 | 290 | 461 | 25 | 382 | 0.5 | 408 | -7.8 | -18.1 | 385 | 3.01 | 665 | 626 | 0.33 |
| | 7 | 7.30 | 293 | 461 | 25 | 409 | 0.6 | 435 | -8.1 | -18.1 | 390 | 2.99 | 761 | 689 | 0.32 |
| | 8 | 7.30 | 279 | 519 | 27 | 461 | 0.7 | 489 | -7.8 | -18.1 | 551 | 3.19 | 1073 | 995 | 0.36 |
| | Mean | 6.86 | 320 | 479 | 27 | 255 | 0.3 | 282 | -8.4 | -16.7 | 303 | 2.51 | 649 | 459 | 0.26 |
| | (SE) | (0.21) | (19) | (12) | (1) | (59) | (0.1) | (58) | (0.5) | (0.6) | (54) | (0.26) | (117) | (111) | (0.06) |
| Stony (Tributary) | 1* | 5.00 | 524 | 451 | 25 | 1 | 0.0 | 26 | -15.6 | -12.3 | 101 | 0.73 | 5 | 26 | 0.00 |
| | 2 | 6.71 | 226 | 501 | 28 | 449 | 0.7 | 478 | -5.0 | -15.9 | 437 | 2.22 | 39568 | 7 | 3.50 |
| | 3 | 7.11 | 148 | 448 | 26 | 338 | 0.4 | 365 | -9.3 | -18.0 | 458 | 3.12 | 10 | 59 | 0.16 |
| | 4 | 6.53 | 245 | 572 | 32 | 375 | 0.4 | 407 | -8.5 | -19.4 | 550 | 3.31 | 704 | 194 | 0.67 |
| | 5 | 7.00 | 479 | 494 | 26 | 601 | 1.3 | 628 | -7.0 | -18.0 | 596 | 2.88 | 1270 | 104 | 0.13 |
| | 6 | 7.37 | 260 | 498 | 27 | 633 | 1.4 | 661 | -8.0 | -18.3 | 1142 | 3.34 | 1936 | 38 | 0.20 |
| | 7 | 7.32 | 230 | 475 | 26 | 570 | 1.2 | 597 | -10.7 | -18.0 | 1078 | 3.56 | 1258 | 227 | 0.60 |
| | Mean | 6.72 | 302 | 491 | 27 | 424 | 0.8 | 452 | -9.1 | -17.1 | 623 | 2.74 | 6393 | 94 | 0.75 |
| | (SE) | (0.31) | (53) | (16) | (1) | (82) | (0.2) | (83) | (1.3) | (0.9) | (140) | (0.37) | (5536) | (32) | (0.47) |

**Table 3.** Characteristics of Stony Creek tributary watersheds (upper panel) and results from the multiple linear
regression model (lower panel). The Stony Creek watershed contained 109 retrogressive thaw slumps (RTS), 92 of
which were in the major tributaries of the Stony Creek mainstem. TR = terrain roughness. EVI = enhanced
vegetation index. Values were used in the multiple linear regression model to determine the drivers of $HCO_3^-$ yields
in Stony Creek tributary watersheds. Model results are shown in the lower panel. Covariates eliminated during
model selection ($RTS_n$, TR, EVI) are not reflected in the lower panel or final model: $ln HCO_3^-$ yield = 1.04$ln$Water
yield + 0.35$ln RTS_{area}$ + 8.76.

| Tributary | $HCO_3^-$ yield ($\mu$M m$^{-2}$ d$^{-1}$) | Water yield (cm d$^{-1}$) | RTS (% area) | RTS ($n$) | Mean TR (m) | Mean EVI |
|---|---|---|---|---|---|---|
| 1 | 2 | 0.20 | 0.00 | 0 | 16.2 | 0.28 |
| 2 | 2819 | 0.61 | 3.50 | 6 | 3.4 | 0.48 |
| 3 | 1047 | 0.18 | 0.16 | 3 | 6.7 | 0.45 |
| 4 | 227 | 0.04 | 0.67 | 50 | 4.5 | 0.45 |
| 5 | 1378 | 0.12 | 0.31 | 11 | 4.7 | 0.46 |
| 6 | 3056 | 0.28 | 0.20 | 8 | 2.7 | 0.47 |
| 7 | 821 | 0.07 | 0.60 | 14 | 3.6 | 0.45 |

| Covariate | Estimate | $t$ | $p$ |
|---|---|---|---|
| $ln$Water yield | 1.04 | 3.6 | 0.02 |
| $ln RTS_{area}$ | 0.35 | 10.5 | < 0.001 |

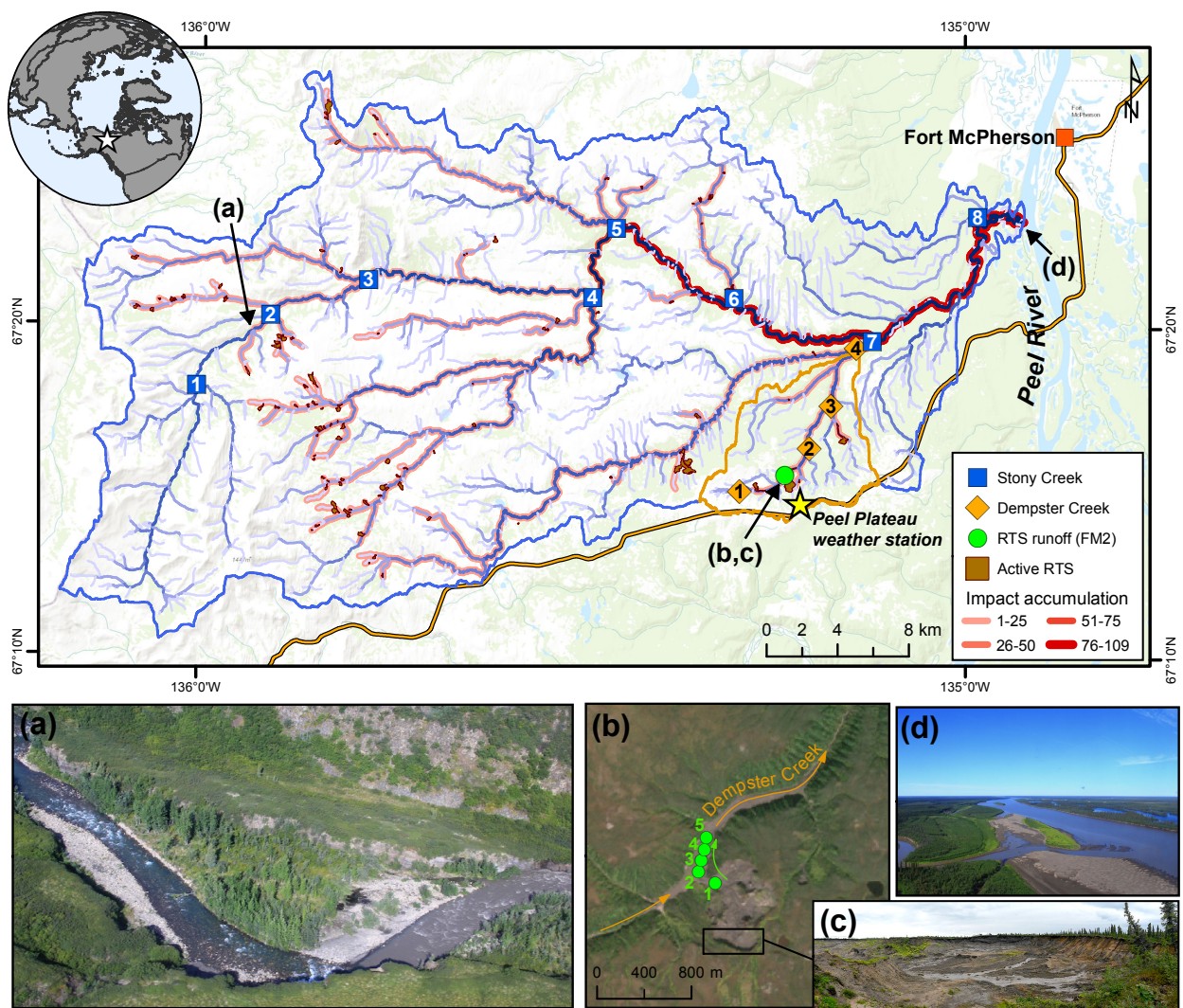

**Figure 1.** Sampling sites on the Peel Plateau (NWT, Canada). Water samples were collected along the mainstem Dempster and Stony Creeks ($n = 12$) and major tributaries ($n = 10$), and from the rill runoff at retrogressive thaw slump (RTS) FM2. Numbers within symbols are sampling sites (Tables 1 and A1). RTS impact accumulation represents the number of active RTSs affecting upstream reaches ($n = 109$) (see Methods Sect. 2.6). (**a**) Aerial photograph of Stony Creek where it was first impacted by RTS activity. (**b**) RTS FM2 runoff transect sampling scheme. RTS FM2 spans ~40 ha, its headwall (**c**) reaches ~25 m in height, and the debris tongue contains $2 \times 10^6$ m$^3$ of sediment (van der Sluijs et al., 2018). (**d**) Aerial photograph of the Stony Creek (lower left) flowing into the Peel River. Satellite image of RTS FM2 in September 2017 (**b**) obtained from Copernicus Sentinel data (European Space Agency, https://sentinel.esa.int/). Basemap: Esri ArcGIS Online © OpenStreetMap contributors, GIS User Community.

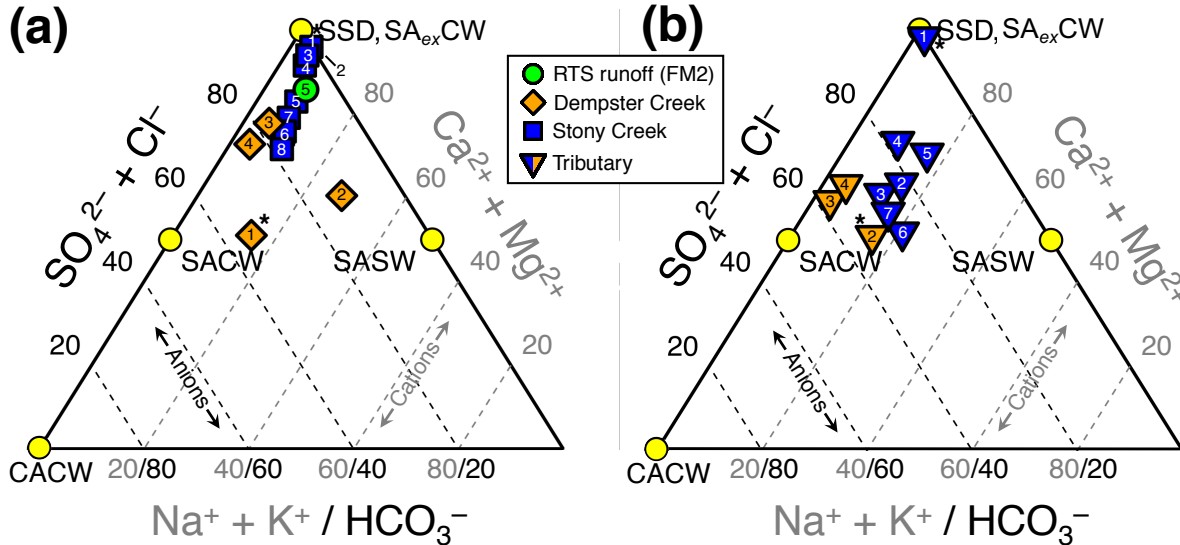

**Figure 2.** Piper diagrams (modified to show the upper half of the diamond plot) showing stream chemistry of the (**a**)
mainstem and retrogressive thaw slump (RTS) FM2 runoff sites and (**b**) tributary sites. Axes and corresponding text
in gray and black reflect the proportions of cations and anions, respectively. Mineral weathering end-members were
derived from the proportional concentration (meq L$^{-1}$) of solutes generated by $H_2CO_3$ carbonate weathering (CACW,
Eq. 1), $H_2SO_4$ carbonate weathering (SACW, Eq. 3), $H_2SO_4$ silicate weathering (SASW, Eq. 4), sulfate salt (e.g.
gypsum) dissolution (SSD, Eq. 5), and carbonate weathering by $H_2SO_4$ in excess (SA$_{ex}$CW, Eq. 7). Site numbers
given within symbols (Table A1). **\*** Not RTS affected.

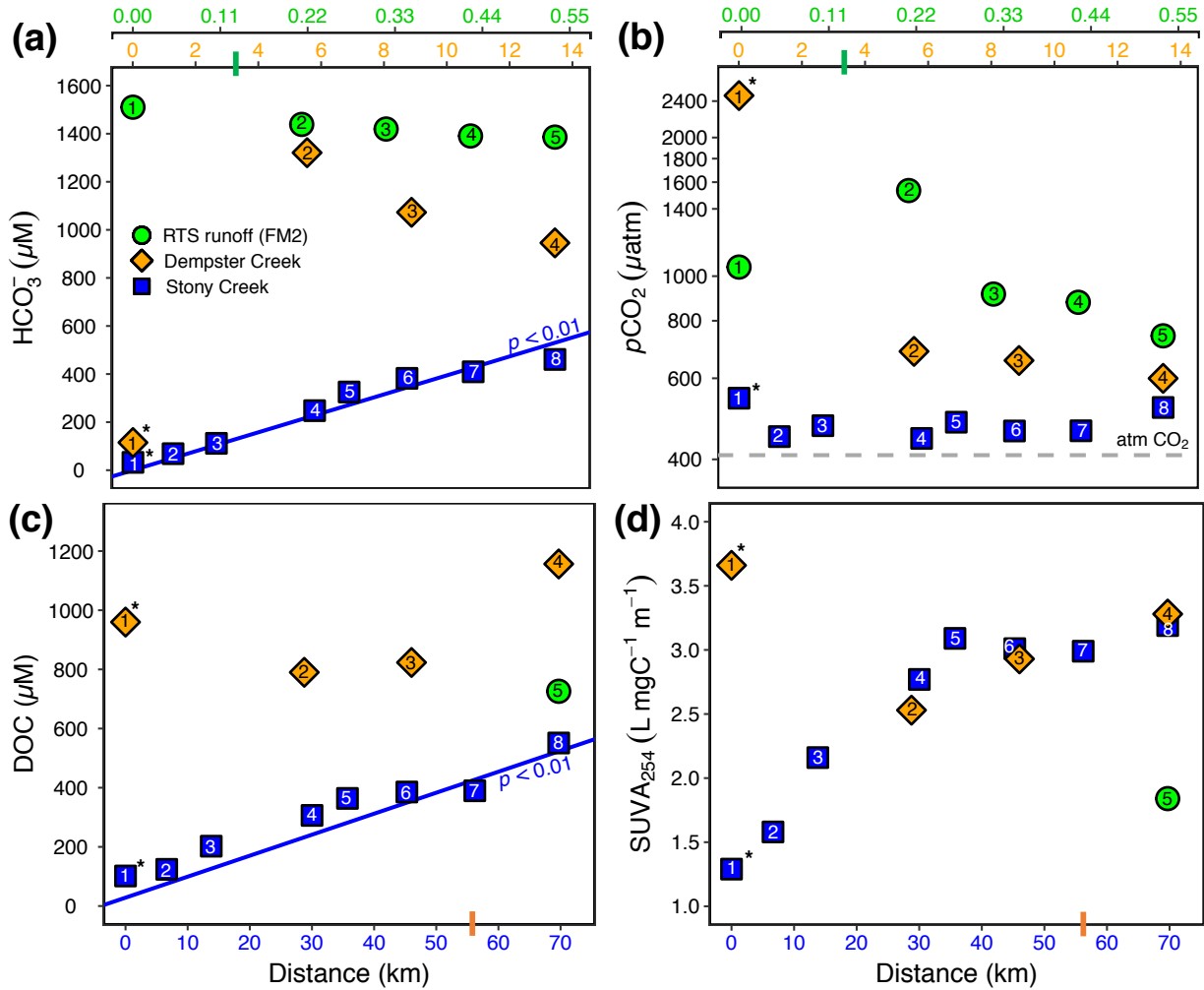

**Figure 3.** (**a**) $HCO_3^-$, (**b**) $pCO_2$, (**c**) dissolved organic carbon (DOC), and (**d**) $SUVA_{254}$ along the RTS FM2 runoff
transect and the mainstem Stony and Dempster Creeks (see locations in Fig. 1). Note different x-axis scales for the
FM2 runoff transect (0–0.55 km, upper x-axis), Dempster Creek (0–14 km, below RTS FM2 x-axis) and Stony
Creek (0–70 km, lower x-axis). For the RTS FM2 runoff, DOC and $SUVA_{254}$ were sampled only at 0.55 km.
Regression lines in (**a**) and (**c**) are from a Mann-Kendall test (details in Sect. 2.7). Bars on x-axes indicate where
RTS FM2 runoff enters the Dempster Creek transect (3.3 km) and where Dempster Creek enters Stony Creek (56
810 km). Site numbers are given within symbols (Table A1). **\***Not RTS affected.

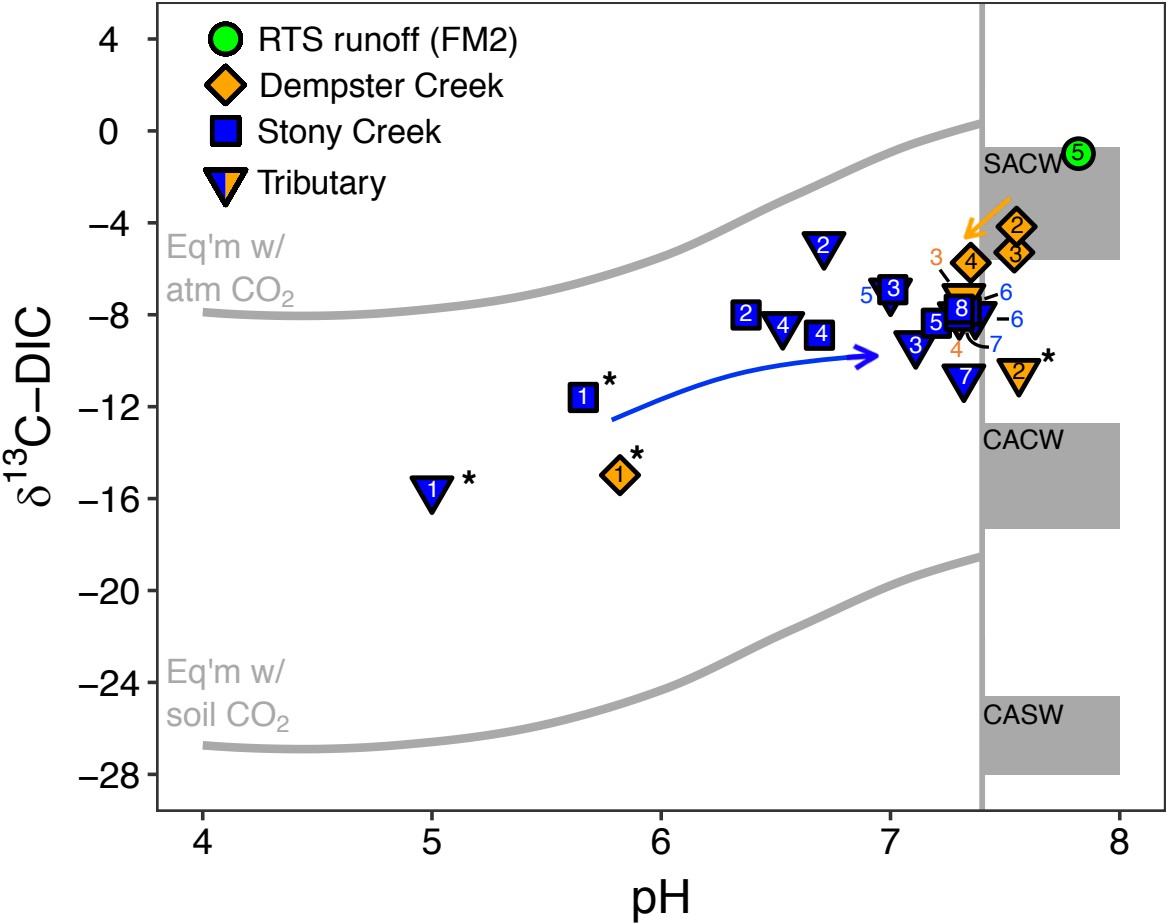

**Figure 4.** The pH and composition of dissolved inorganic carbon stable isotopes ($\delta^{13}$C-DIC) in streams. The upper
and lower reference lines depict theoretical end-members for equilibrium reactions (mixing with atmospheric and
biotic $CO_2$, respectively). Gray boxes span theoretical end-member values for kinetically controlled mineral
weathering reactions (SACW = $H_2SO_4$ carbonate weathering, CACW = $H_2CO_3$ carbonate weathering, CASW =
$H_2CO_3$ silicate weathering) (see Sect. 2.4 for derivation of end-members). The vertical line corresponds to the pH at
which $\geq$ 90% of DIC is $HCO_3^-$, for the mean observed stream water temperature (11.7°C). At pH < 7.4, $\delta^{13}$C-DIC
values primarily reflect equilibrium (rather than kinetic) controls on DIC cycling. Arrows reflect increasing
downstream distance from the headwaters in Stony Creek and from the first retrogressive thaw slump (RTS)
affected site in Dempster Creek. Site numbers given within symbols (Table A1). **\*Not RTS affected.**

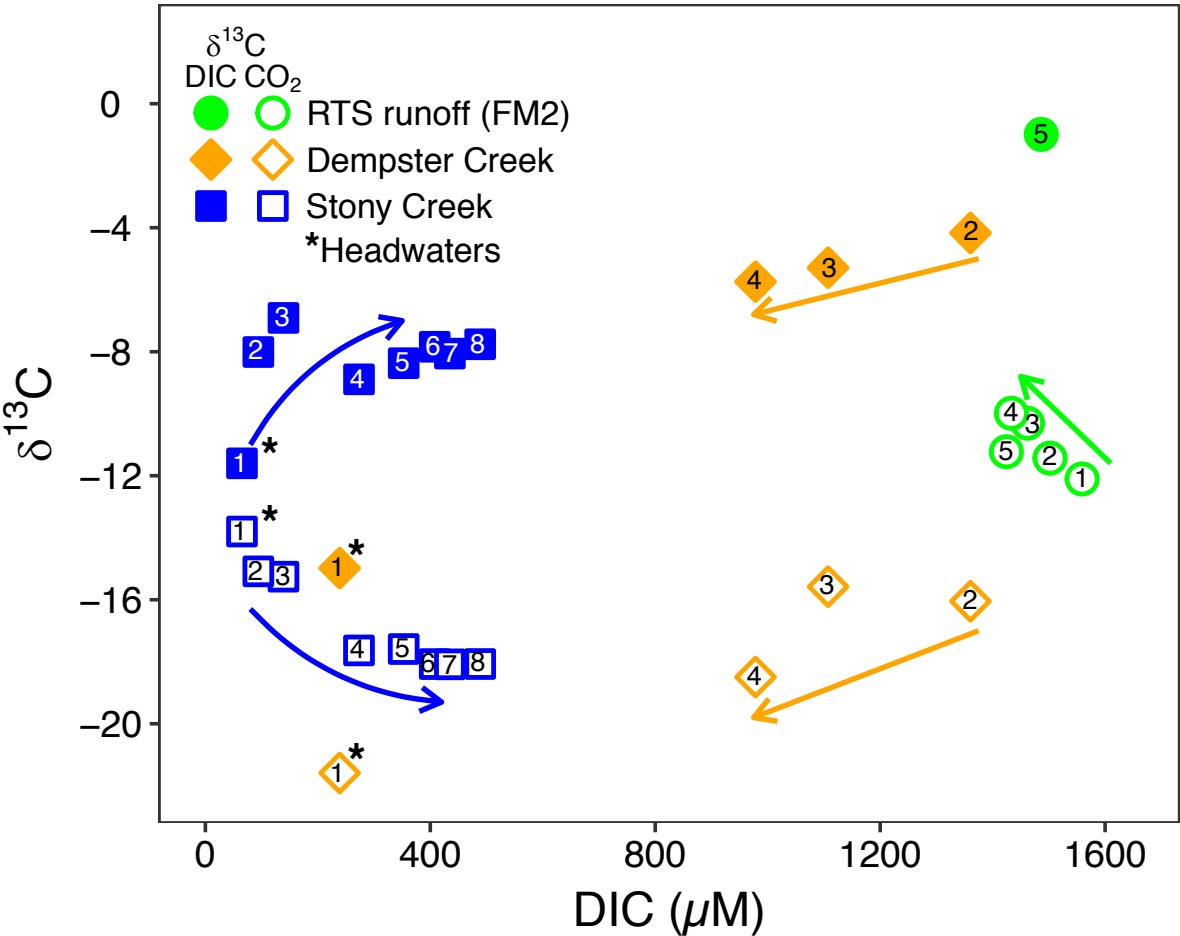

**Figure 5.** The composition of DIC and $CO_2$ stable isotopes at varying DIC concentrations along the Dempster and
823 Stony Creek mainstems, and in the rill runoff of retrogressive thaw slump (RTS) FM2. Arrows reflect increasing
downstream distance from headwaters in Stony Creek, from the first RTS-affected site in Dempster Creek, and from
the start of the FM2 runoff transect. Site numbers given within symbols (Table A1). *Site in headwaters and not
affected by RTSs.

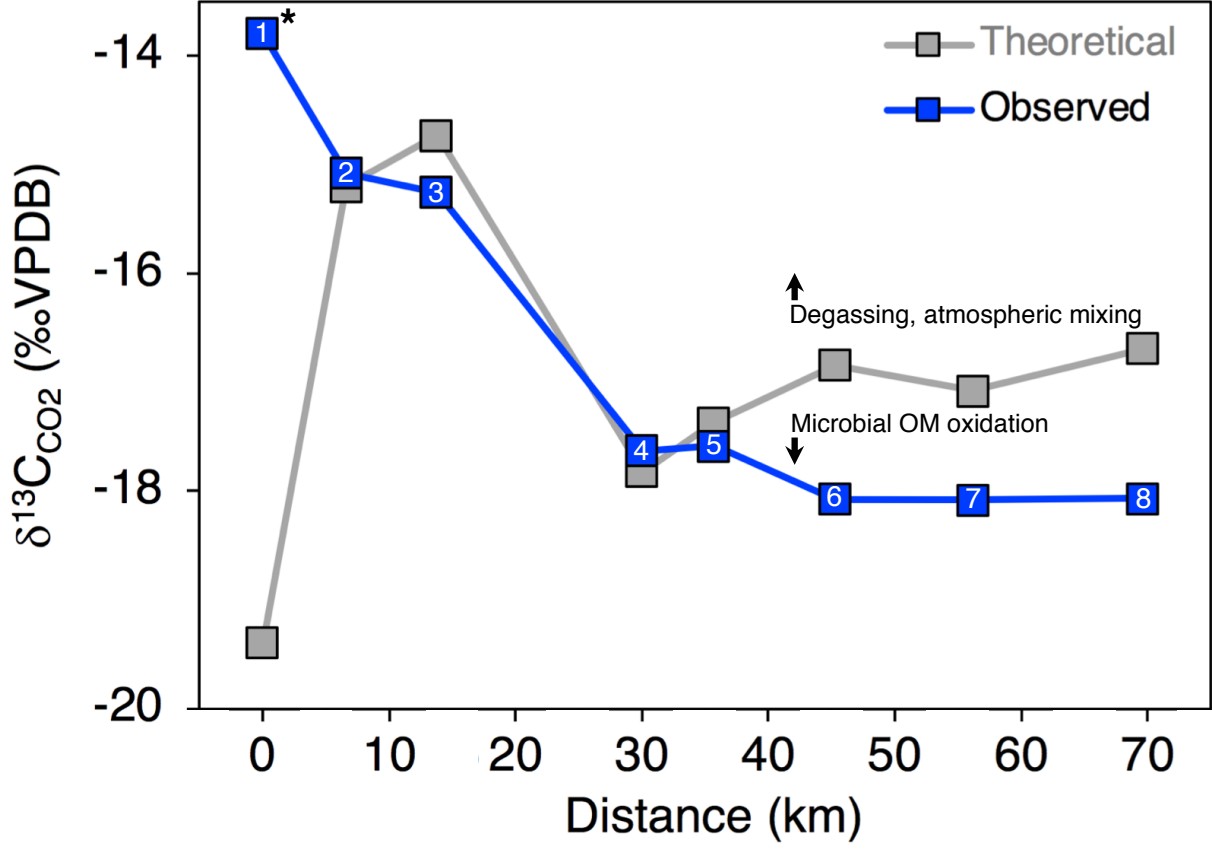

**Figure 6.** Observed and expected $\delta^{13}$C-CO$_2$ values along the Stony Creek mainstem. Theoretical $\delta^{13}$C-CO$_2$ values
were calculated as detailed in Sect. 2.6 and reflect changes in CO$_2$ due to DIC speciation (i.e. H$_2$CO$_3$ ⇌ H$^+$ + HCO$_3^-$,
Eq. 8). Deviation from theoretical $\delta^{13}$C-CO$_2$ values by observed values thus indicates isotopic effects from
degassing and/or microbial oxidation of organic matter (OM), as indicated by the arrows. Site numbers given within
symbols (Table A1). *Site was not affected by retrogressive thaw slumps.

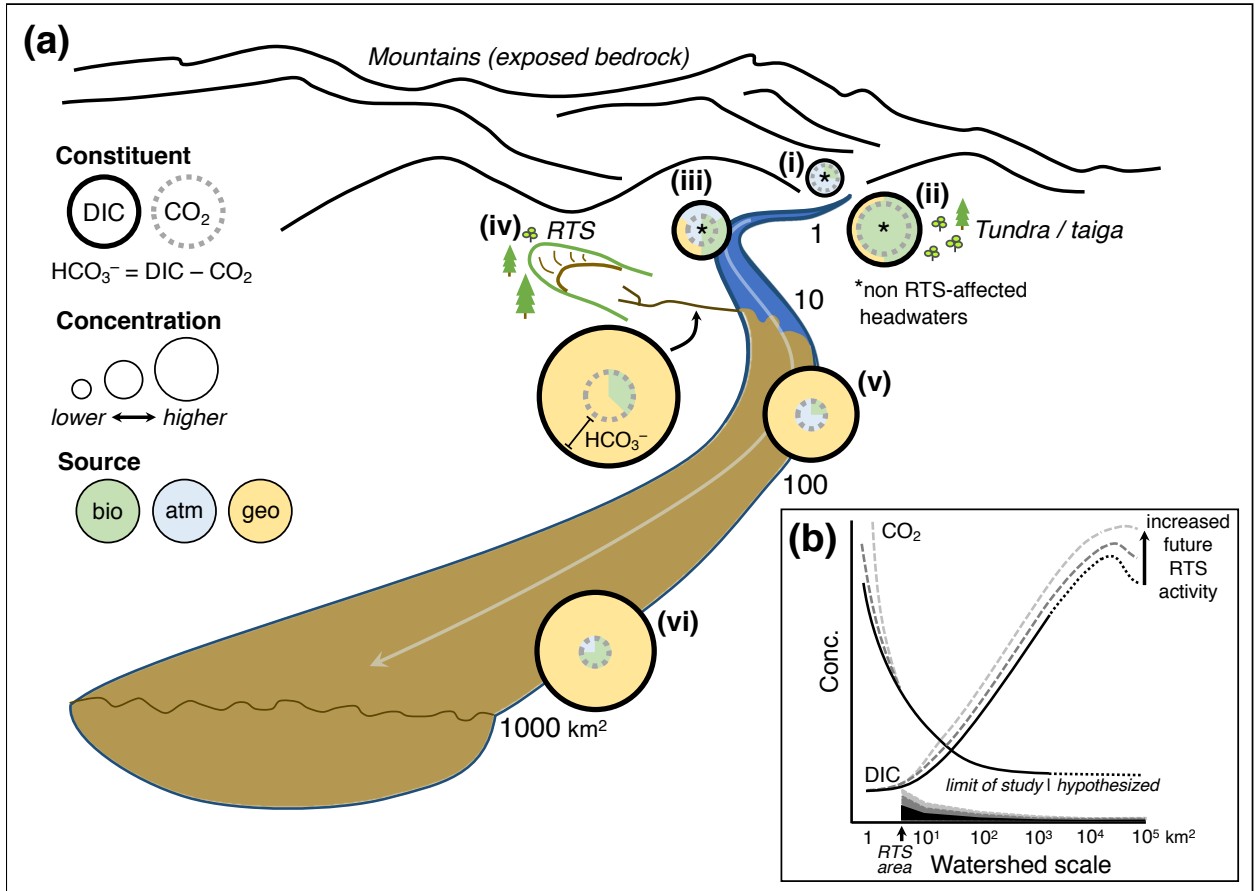

**Figure 7.** (**a**) Conceptual model of retrogressive thaw slump (RTS) activity and mineral weathering effects on carbon cycling in glaciated thermokarst terrains like the Peel Plateau. Source abbreviations: bio = biogenic, atm = atmospheric, geo = geogenic. (**b**) RTS effects on $CO_2$ and DIC ($\Sigma[CO_2$, carbonate alkalinity]) observed in this study (dark solid line), projected across broader scales in the modern-day (dark dotted line), and under hypothetical future scenarios of increasing RTS activity (medium- and light-gray dashed lines). Shaded regions along x-axis depict relative RTS area approximated for modern-day (black) and for hypothetical future increases in RTS area (medium- and light-gray).

# Appendices

**Table A1.** Sampling site characteristics. Retrogressive thaw slump (RTS) FM2 runoff was a tributary to Dempster Creek (confluence upstream of site 2) and Dempster Creek was a tributary to Stony Creek (confluence upstream of site 8). Coordinates reported in decimal degrees. *Not RTS affected.

| Creek | Site | Type | Latitude (DD) | Longitude (DD) | Sampling date | Distance (km) | Elevation (m) | Stream order (Strahler) |
|---|---|---|---|---|---|---|---|---|
| RTS FM2 | 1 | Runoff | 67.25639 | -135.23422 | 7/31/17 | 0 | – | 1 |
| RTS FM2 | 2 | Runoff | 67.25726 | -135.23756 | 7/31/17 | 0.22 | – | 1 |
| RTS FM2 | 3 | Runoff | 67.25813 | -135.23700 | 7/31/17 | 0.33 | – | 1 |
| RTS FM2 | 4 | Runoff | 67.25894 | -135.23636 | 7/31/17 | 0.44 | – | 1 |
| RTS FM2 | 5 | Runoff | 67.25986 | -135.23595 | 7/31/17 | 0.55 | – | 1 |
| RTS FM2 | 5 | Runoff | 67.25981 | -135.23587 | 7/30/17 | – | 271 | 1 |
| Dempster | 1* | Mainstem | 67.25181 | -135.29456 | 7/31/17 | 0 | 407 | 3 |
| Dempster | 2 | Mainstem | 67.27364 | -135.20409 | 7/29/17 | 5.6 | 194 | 4 |
| Dempster | 3 | Mainstem | 67.29500 | -135.17570 | 7/27/17 | 8.9 | 132 | 4 |
| Dempster | 4 | Mainstem | 67.32336 | -135.14133 | 7/27/17 | 13.5 | 67 | 4 |
| Dempster | 2* | Tributary | 67.27364 | -135.20367 | 7/29/17 | – | – | 2 |
| Dempster | 3 | Tributary | 67.29497 | -135.17538 | 7/27/17 | – | – | 3 |
| Dempster | 4 | Tributary | 67.32414 | -135.14252 | 7/27/17 | – | – | 4 |
| Stony | 1* | Mainstem | 67.30280 | -136.00468 | 7/27/17 | 0 | 575 | 4 |
| Stony | 2 | Mainstem | 67.33878 | -135.90912 | 7/27/17 | 6.6 | 474 | 4 |
| Stony | 3 | Mainstem | 67.35704 | -135.78165 | 7/25/17 | 13.8 | 382 | 4 |
| Stony | 4 | Mainstem | 67.34913 | -135.48802 | 7/25/17 | 30.0 | 230 | 5 |
| Stony | 5 | Mainstem | 67.38380 | -135.45747 | 7/25/17 | 35.7 | 184 | 6 |
| Stony | 6 | Mainstem | 67.34879 | -135.30302 | 7/25/17 | 45.3 | 123 | 6 |
| Stony | 7 | Mainstem | 67.32732 | -135.12160 | 7/25/17 | 56.2 | 57 | 6 |
| Stony | 8 | Mainstem | 67.39000 | -134.98380 | 7/25/17 | 69.7 | 6 | 6 |
| Stony | 1* | Tributary | 67.30367 | -136.00421 | 7/27/17 | – | – | 3 |
| Stony | 2 | Tributary | 67.33933 | -135.90836 | 7/27/17 | – | – | 3 |
| Stony | 3 | Tributary | 67.35719 | -135.78311 | 7/25/17 | – | – | 4 |
| Stony | 4 | Tributary | 67.34860 | -135.48773 | 7/25/17 | – | – | 5 |
| Stony | 5 | Tributary | 67.38467 | -135.45607 | 7/25/17 | – | – | 4 |
| Stony | 6 | Tributary | 67.34882 | -135.30196 | 7/25/17 | – | – | 4 |
| Stony | 7 | Tributary | 67.32703 | -135.12213 | 7/25/17 | – | – | 5 |

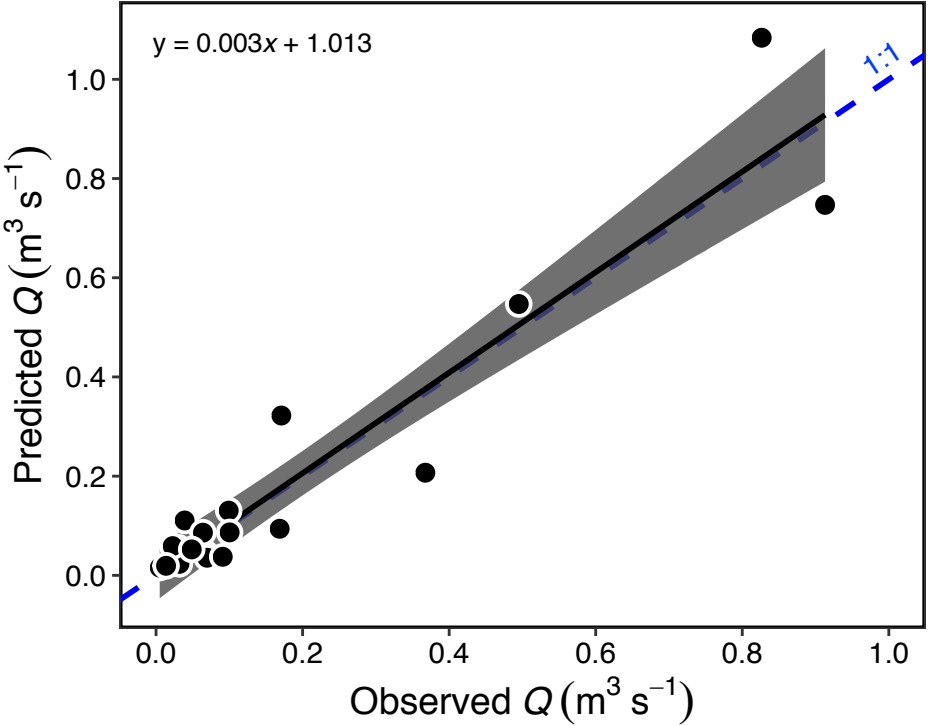

**Figure A1.** Estimated vs. measured discharge ($Q$) ($p < 0.001$, $R^2 = 0.89$, $F_{1,18} = 150$) for 20 streams in the Stony
Creek watershed. Grey band represents the 95% confidence interval shown around the regression. Estimates were
made using measurements of stream width, $Q$, and a hydraulic geometry model (Gordon et al., 2004) (see Sect. 2.6).
The model (Eq. 1) was used to estimate $Q$ in the Stony Creek tributaries.

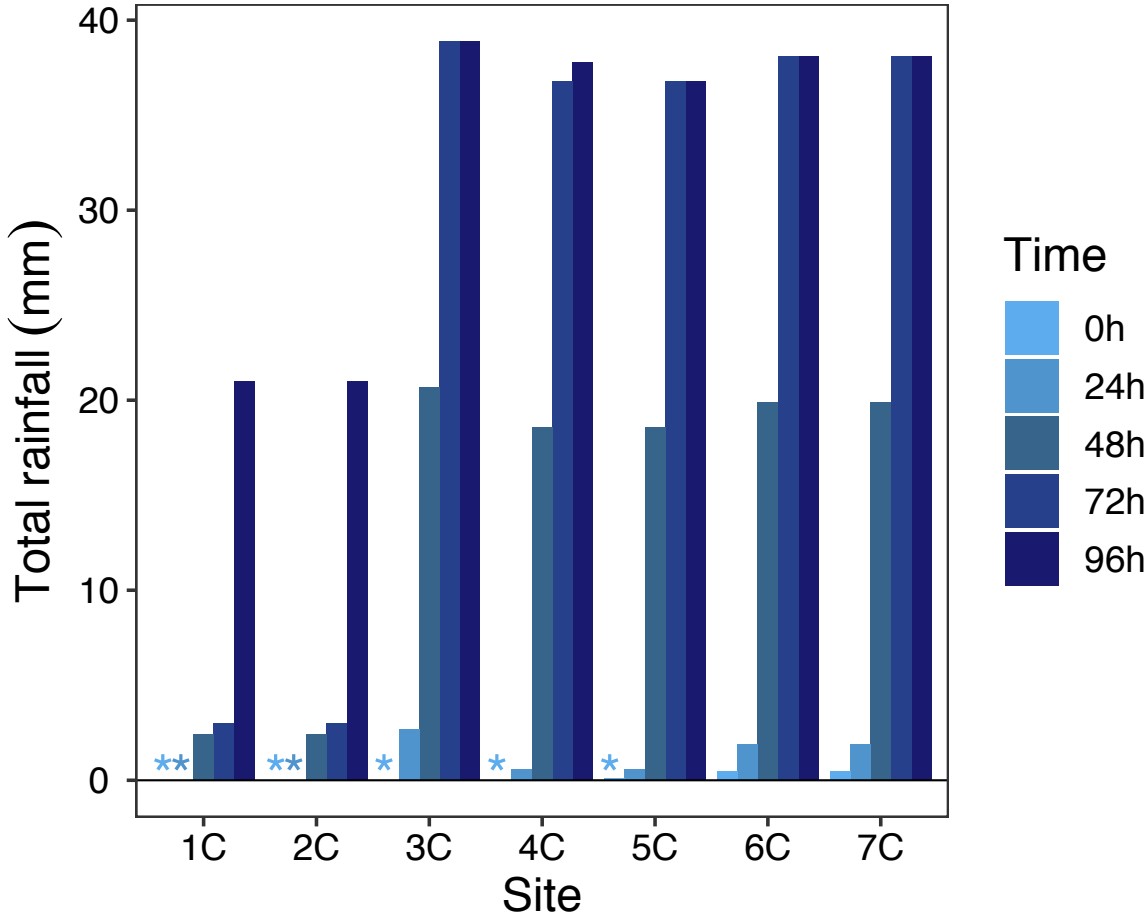

**Figure A2.** Total rainfall in 24 h increments preceding the sampling of each Stony Creek tributary. Rainfall data
were obtained from a Government of Northwest Territories weather station on the Peel Plateau located near the RTS
FM2. Locations of tributary sampling sites and the weather station are shown in Fig. 1. **\***Indicates no rainfall in the
24 h window.