# Peer review of "Thermokarst amplifies fluvial inorganic carbon cycling and export across watershed scales on the Peel Plateau, Canada"

_Biogeosciences, 2020_

## Referee Comment (RC1) · Anonymous Referee #1 · 23 Apr 2020

Zolkos et al., present a high quality characterisation of running water chemistry in a sub-catchment of the Peel River where the aim was to determine the effect of retrogressive thaw slump (RTS) on DIC sources and export. The study design, incorporating three transects at different spatial scales (1. retrogressive thaw slump (RTS) runoff water, 2. an intermediate size catchment with direct fed with RTS runoff, 3. A large catchment fed indirectly with RTS runoff through its tributaries), is an interesting and innovative sampling approach. The dataset, including a large number of key variables, is also of very high quality. The research question is also highly relevant to our understanding of the permafrost climate-feedback. The author rightfully stated that changes in carbonate alkalinity export in response to permafrost degradation has been far less

studied that those for organic carbon and carbon dioxide. The study is also taking place in a relatively understudied region, which makes it even more valuable. While the subject and design of the study is of high quality, I find that the discussion and conceptualisation of the result need significant improvement. In short, the paper does not make full use of its potential.

The novelty of the paper lies in the approach of scales on the effect of RTSs on DIC export. At the moment, these three scales are taken separately, into three almost individual studies. Is the message simply that we can perceive the RTS effect at each of those three scales or is there a greater interpretation of how these effects integrate with increasing spatial scale and decreasing land-water connectivity? The paper would have more impact if the author could conceptualise these results and formulate how RTSs affect DIC cycling across scales rather than simply testifying that it has an influence. For example, does the "RTS effect" amplifies, is conserved, accumulates or becomes diluted with increasing scale. The author already documented that RTSs alter riverine DIC cycling in a previous publication in GRL (Zolkos et al, 2018). At present, this manuscript adds little to this state of knowledge, but this could be remediated by conceptualising further the effect of scales.

I have made a few suggestions to improve the presentation of results and conceptualisation of the discussion.

The sampling design is interesting and valuable, but arguably difficult to communicate to the reader. Having the Results and Discussion section together makes it even harder for the reader to put together the key results, and follow the discussion points that are mixed through the text. I recommend to separate these two parts. The first part of the results and discussion section details the changes in water chemistry in each of the three transect. The first part of the Result and Discussion section (section 3.1, 3.2 ad 3.3) details the water chemistry patterns in each transect with discussion points mixed through the text. Having this structure increases the impression that this study actually involves three separate studies rather than one. I suggest to also structure the

results by water chemistry variable rather than sites. The study measured a large number of important and interesting water chemistry variables. Each should be presented clearly in the result section for the reader to identify. Each sections should provide, among other things, the overall range in values for the whole study, compare this range between each of the three scales and within each transect.

Rather than naming the sites by their official river name, why not call them with a more conceptual name that represents the idea behind the sampling design. I like that the symbols in figures have numbers to indicate their position along the transect, but RTS FM2 site DC and SC has little meaning for the reader. I have also provided a few suggestions below to format the figures in a more visually telling way.

Playing with your dataset I attempted to trace the d13CO2 source with the miller-trans plots, I found a clear difference in that value for the RTS runoff site (d13Csource -12‰ and the DC and SC rivers (-22.9‰. This suggest two predominant CO2 sources in this catchment and those end-members could potentially be used for calculations. The RTS site is consistent with a geogenic CO2 source, while the rivers have a predominant biogenic soil CO2 source. Would it be worth including this kind of approach to your results?

The effect of scales, with a varing degree of terrestrial connectivity, is only discussed in the context of circumpolar region with, but I believe that the study should also be put in the context of the broader literature, including lower latitude catchments, where many studies have also examined the effect of changing land-water connectivity with size.

I find it interesting to see the downstream changes in HCO3 concentration. It could be worth mentioning that studies modelling stream CO2 evasion based on d13CO2 value assume that carbonate alkalinity is conserve in river networks (Polsenaere 2012 Geochimica et Cosmochimica Acta and Venkiteswaran 2014 PLOS one).

The term thermokarst and retrogressive thaw slumps are used interchangeably, I suggest to stick to retrogressive thaw slumps since this was the focus of the study and the

findings may not be applicable to all kinds of thermokarst disturbances.

The term "abiotic-inorganic processes" is a vague term to me, what are the processes included in that? Carbonate equilibrium reaction and CO2 evasion? Could you be more specific? Either a more specific term should be used or the choice of terminology should be justified in the text.

Abstract: Line 21:26: The results/discussion section of the abstract list changes in water chemistry in each of the three transect. I believe this section, and other relevant section in the main text, should come up with a more unified message of the RTS effect across scales, rather than at individual scales.

Line 30: I have trouble with the word "prevalent" here since it implies that one is larger than the other, while such calculations have not been done in the study.

Introduction: Lines 32-33: This first sentence would introduce better the second paragraph where the source/sink relationship of DIC is detailed. This first paragraph discusses how the arctic landscape is changing and what is known of its influence on DIC export.

Lines: 41-44: Since we include CO2 in this pool, the increase in soil respiration and/or increase in aquatic DOC degradation should also be part of this list. The citations be separated to indicate which if the listed mechanisms have been highlighted by each study.

Lines 47-64: This is mostly textbook material and could be synthesised. I think what you want to express is why it's important not only to account for the mass of DIC exported but also to identify its sources DIC. Without this we can't determine whether this is a new sink or source of C in the short and long term perspective. I would suggest to move some of this information in section 2.5 of the methods and rework this paragraph to emphasise the simply the importance of source separation for DIC.

Line 65: Could you state the representativeness of this study, which areas across the

circumpolar north could be similar to your studied location?

Line 94: This is a very important point which I think should be stated earlier in the introduction and given more emphasis.

Methods Line 133: How deep are those rivers? Can stream order be provided somewhere too.

Section 2.3 This is a nice way to work around flow measurement scarcity in this region. But should this section come after section 2.6?

Line 212: What threshold was used in the flow accumulation to delineate the stream lines and catchment boundary? Was it validated with the areal photos mentioned on line 155 or something else?

Section 2.4, shouldn't this section be called water chemistry analysis?

Line 245: Why not model the full carbonate alkalinity pool (HCO3+ CO3)? Arguably the CO3 pool is small at this pH range and shouldn't make a much difference to the model, but I find this conceptually more appropriate.

Results and Discussion

Section 3.1, 3.2 and 3.4 starts with a sentence stating how the author interprets DIC sources and cycling in each transect. This seems odd to me. I would rather the author takes me to that conclusion by presenting the results first.

Line 275: The Miller-tans analysis supports that as well. Again, this statement comes before presenting the key results.

Line 290: Is the term "geogenic" more appropriate?

Line 285: Probably right, but do you have any measurements or estimates of CH4 concentration at the source - in the groundwater? The CH4 might have evaded already, but its imprint on d13CO2 values might still be there.

Line 301: That is also supported with the Miller-tans. But this biogenic soil CO2 source seems to prevail in other sites as well albeit with some mixing with the geogenic source. Could mixing between the H2SO4 weathering and biogenic soil CO2 be back calculated?

Line 314: This is also demonstrated in headwater streams at other latitudes and should be mentioned as well.

Line 314; Again I find this term "abiotic inorganic" to be vague. And what do you mean by dominates? Abiotic dominates biotic, or inorganic dominates organic?

Line 351: What does "amplified inorganic carbon cycling" means?"

Line 371: The term biotic is used here, but could the DOC be degraded photochemically as well?

Line 380: Is the model intended only to bring evidence to the fact that RTS increase alkalinity export or for a possibly larger modelling/budget exercises? Could you make use of that model already in this paper for a final "wrap up" exercise ?

Line 393: Dominate over what?

Line 405: This "conceptual model of land-freshwater linkages" needs more elaboration. As it stands, this model seems more like a list of DIC sources and sinks across this catchment than a generalisable model. A starting point would be to determine how does it integrates with other models at lower latitudes? How do changes in DIC sources and sinks caused by RTS integrates with other water chemistry properties and C species (organic vs inorganic) that were documented in other studies?

Line 410: What exactly was "striking"?

Line 414-418: How does this conclusion differ from the authors previous publication cited here?

Line 432: Should your model be used to that effect? If so, it should be stated.

Line 443: " change for C cycle in the rapidly changing arctic landscape"

Authors Contribution What did D.O. do?

Figures and Tables

Table2: Should these values be presented in supplementary and only the model be presented in this table? I find the second part of the table easy to miss.

Figure 1: The map feels quite dense, can the photos and context map be placed outside? Could the bedrock lithologies be illustrated on the map?

Figure 2: The figures should be placed vertically rather than horizontally since they have the same x-axis. Also in caption, please clearly state that the top x-axis is for the dempster creek transect while the second is for the stony creek. Could the points for each transect be connected with a line for visualisation. Could $HCO_3$ and $CO_2$ concentration be on the same unit? Could there be a third 3 axis for the distance along the RTS runoff transect ?

Figure 4: Can you give a reference to these end-members

Figure 5: Open vs closed symbols would be clearer perhaps?

Figure 6 and 7: Should these two figures be merged with Figure 2? This would help draw a more complete picture of simultaneous changes in water chemistry along the transects. Why isn't there a similar figure for d13C-DIC values?

Figure 8: This is a nice schematic, but it limits the scope of the study. The schematic mostly lists the sources and transformation of DIC in this catchment. Does it only applies to this catchment, i.e. was the goal to map those processes, or can it be generalised to other catchments? I think this figure could be useful if it was to conceptualise the effect of RTS across scales, not just make a summary of all the processes identified in the data for this specific catchment. I have in mind something along the lines of Hotchkiss et al. 2015 NatGeo Figure 3.

Table A1. Why not keep the distance units the same and just add decimals for FM2 site.

Table A2: I find this could be useful in the main manuscript since it also provides a list of the DIC sources you are trying to separate.

―――――――――――――――――

---

## Referee Comment (RC2) · Anonymous Referee #2 · 26 Apr 2020

Review on MS# BG-2020-111 "Thermokarst amplifies fluvial inorganic carbon cycling and export across watershed scales on the Peel Plateau, Canada" by Scott Zolkos et al.

Zolkos et al. present a detailed and high quality characterisation of running water chemistry in a sub-catchment of the Peel River. This work was to determine the effect of retrogressive thaw slump (RTS) on DIC sources and export. The research design, incorporating three transects at different spatial scales, is an interesting sampling approach. The dataset, including a large number of key variables, is also of very high quality. The research question is highly relevant to our understanding of the permafrost

climate-feedback.

While the design of the study is of high quality, I find that the discussion of the results needs some improvement.

I think the influence of thermokarst on fluvial inorganic carbon cycling and export is reflected in two aspects. One is the change in runoff, and the other is the change in DIC concentrations and sources. The authors have discussed the latter more clearly, but the former needs to be done further. In addition, the authors used the change in concentration and isotope of DIC to indicate the sulfuric acid carbonate weathering, but the sulfuric isotopic evidence may be the more direct one. Could they add this to further strength their conclusions?

---

## Referee Comment (RC3) · Anonymous Referee #3 · 5 May 2020

Overview:

In this study the authors investigate how permafrost thaw affects mineral weathering sources of inorganic carbon (IC), and how the fluvial IC is cycled across different scales. Specific focus is on retrogressive thaw slumps (RTS) and their major contribution to IC yields and biogeochemical processes across fluvial networks draining permafrost regions. The study is based on one synoptic summer sampling campaign of three different fluvial transects covering different scales, and where samples were taken for a comprehensive set of chemical and isotopic variables. The authors conclude that rapid weathering in the RTS runoff enhance both atmospheric CO2 emission and downstream DIC transport. They further show that the IC signal from RTS have a major downstream impact across large scales although the RTS impacted area covered less than a 1% of the total catchment area.

The manuscript focus on an important topic that is very suitable for publication in Biogeosciences. The current thaw of permafrost regions is of major concern and the response in the landscape C cycling is a central issue. Much of the literature is focusing on the mobilization of organic C stocks and the subsequent mineralization into CO2 and CH4. In comparison, relatively little focus is given to the inorganic C mobilization and to what degree mineral weathering upon permafrost act as a source or sink for atmospheric C, and how it affects biogeochemical processes in aquatic systems.

General comments:

With this background the manuscript is an important contribution to the research field. The authors present a comprehensive and neat data set from a data scarce region, and where they disentangle different sources and processes affecting the fluvial IC in a (mostly) very convincing way. The manuscript is very well written but I have some points that need to be clarified prior to a publication. These issues are mostly to strengthen the argumentation by the authors but also to fully capitalize on their findings.

Detailed comments:

Ln 15-18, a very long sentence with plenty of information. I suggest to split it.

Ln 153-160, it is hard to grasp the uncertainty of the stream flow section. i.e. how certain the Q estimates are. On the other hand, the water or solute yields are a relatively minor part of the ms.

Ln 237-239, how come these three variables were used in the MLR? Comes currently a bit out of the blue and needs to be better motivated.

Ln 239-245, again it is hard to judge the certainty in this modelling effort given the already above raised concern about the Q estimation.

Ln 259-, I guess very much a question of personal taste but I feel the ms do not benefit from the mixing of results and discussion. It would be easier to keep focus by separating them in my opinion.

Ln 278, I am not familiar with the given reference, but what is meant by "regional carbonate"? Also in this couple of sentences, I agree with the overall argumentation, but can you completely rule out a biotic source contribution? The fractionation between carbonate and CO2 (8‰ is rather theoretical. Could a mixing with geogenic and biogenic IC be possible for generating 13C-CO2 of -11.4 to 12.1‰ You have a substantial DOC pool which is also cited by being "relatively biolabile".

Ln 285, how CH4 was sampled is mentioned in the methods but from what I see this is the only place where any data is presented, and then very shortly. Maybe the data is saved for another story but I believe it would further strengthen the story if it could be included for example in table 1 and with subsequent incorporation in the text.

Ln 310-313, yes it could be due to adsorption to RTS sediments, but I guess it could also be due to lower mineralization than degassing rates. Might be worth to mention.

Ln 347-349, is it really clear that biotic CO2 were the primary source of DIC in the headwaters of Stony Creek? Could not geogenic sources still be highly influential? The 13C-DIC and 13C-CO2 values (-11.6 and -13.8‰ respectively) points towards a biogenic/geogenic mixing, or?

Ln 403-405, do the study really evaluate "across gradients of thermokarst disturbance"? I believe something like influence of RTS on IC cycling and how this signal is propagated across different fluvial scales is better describing the story.

Ln 419-434, I somehow miss the full interpretation of the findings of the current study for the large scale picture. How do you suggest your results should be considered in large scale estimates, i.e. how does it affect the previous judgement of the area as a "modest source of CO2".

A general question: how common are RTS across permafrost regions worldwide? How applicable are the findings here for other areas?

Figure 1. For a non-north American reader, a more large-scale inset of where the area is found would be appreciated.

---

## Author Comment (AC1) · 12 Jul 2020

**Anonymous Reviewer #2**

Zolkos et al. present a detailed and high quality characterisation of running water chemistry in a sub-catchment of the Peel River. This work was to determine the effect of retrogressive thaw slump (RTS) on DIC sources and export. The research design, incorporating three transects at different spatial scales, is an interesting sampling approach. The dataset, including a large number of key variables, is also of very high quality. The research question is highly relevant to our understanding of the permafrost climate-feedback.

*Thank you very much for the thoughtful and helpful comments.*

While the design of the study is of high quality, I find that the discussion of the results needs some improvement.

I think the influence of thermokarst on fluvial inorganic carbon cycling and export is reflected in two aspects. One is the change in runoff, and the other is the change in DIC concentrations and sources. The authors have discussed the latter more clearly, but the former needs to be done further. In addition, the authors used the change in concentration and isotope of DIC to indicate the sulfuric acid carbonate weathering, but the sulfuric isotopic evidence may be the more direct one. Could they add this to further strength their conclusions?

*We agree that thermokarst influences inorganic carbon cycling via changes in DIC concentrations and sources (e.g. Zolkos et al. 2018). However, discharge within thaw slumps is relatively small compared to the streams affected by slumps, so changes to runoff associated with slumping are likely to be negligible, yet we lack direct evidence for such an assertion. Also, thank you for the suggestion to consider sulfur isotopes as evidence of $H_2SO_4$ carbonate weathering. The sulfur isotopes we measured and reported in our 2018 GRL paper (Zolkos et al. 2018) do align with sulfate derived from sulfide oxidation. We now briefly consider this in our revised Discussion.*

*Zolkos, S., Tank, S. E., & Kokelj, S. V. (2018). Mineral Weathering and the Permafrost Carbon‑Climate Feedback. Geophysical Research Letters, 45(18), 9623-9632.*

---

## Author Comment (AC2) · 12 Jul 2020

**Anonymous Reviewer #3**

In this study the authors investigate how permafrost thaw affects mineral weathering sources of inorganic carbon (IC), and how the fluvial IC is cycled across different scales. Specific focus is on retrogressive thaw slumps (RTS) and their major contribution to IC yields and biogeochemical processes across fluvial networks draining permafrost regions. The study is based on one synoptic summer sampling campaign of three different fluvial transects covering different scales, and where samples were taken for a comprehensive set of chemical and isotopic variables. The authors conclude that rapid weathering in the RTS runoff enhance both atmospheric CO2 emission and downstream DIC transport. They further show that the IC signal from RTS have a major downstream impact across large scales although the RTS impacted area covered less than a 1% of the total catchment area.

The manuscript focus on an important topic that is very suitable for publication in Bio-geosciences. The current thaw of permafrost regions is of major concern and the response in the landscape C cycling is a central issue. Much of the literature is focusing on the mobilization of organic C stocks and the subsequent mineralization into CO2 and CH4. In comparison, relatively little focus is given to the inorganic C mobilization and to what degree mineral weathering upon permafrost act as a source or sink for atmospheric C, and how it affects biogeochemical processes in aquatic systems.

*Thank you for the encouraging comments. We appreciate it!*

General comments:

With this background the manuscript is an important contribution to the research field. The authors present a comprehensive and neat data set from a data scarce region, and where they disentangle different sources and processes affecting the fluvial IC in a (mostly) very convincing way. The manuscript is very well written but I have some points that need to be clarified prior to a publication. These issues are mostly to strengthen the argumentation by the authors but also to fully capitalize on their findings.

*Many thanks for your helpful feedback. Please find our replies below.*

Detailed comments:

Ln 15-18, a very long sentence with plenty of information. I suggest to split it.

*Revised.*

Ln 153-160, it is hard to grasp the uncertainty of the stream flow section. i.e. how certain the Q estimates are. On the other hand, the water or solute yields are a relatively minor part of the ms.

*This is a fair point. We added a figure to the appendix which shows the relationship between our discharge measurements and the estimates from our model. As the figure shows, the 95% confidence interval is relatively larger at higher discharge levels. The strong, linear relationship provides some confidence in our estimates of discharge.*

[Figure]

**Figure A2.** *Estimated vs. measured discharge (Q) (p < 0.001, $R^2$ = 0.89, $F_{1,18}$ = 150) for 20 streams in the Stony Creek watershed. Grey band represents the 95% confidence interval around the regression. Estimates were made using measurements of stream width, Q, and a hydraulic geometry model (Gordon et al. 2004) (see Sec. 2.6). The model (Eq. 1) was used to estimate Q in the Stony Creek tributaries.*

*Gordon, N. D., McMahon, T. A., Finlayson, B. L., Gippel, C. J., & Nathan, R. J. (2004). Stream hydrology: an introduction for ecologists. John Wiley and Sons.*

Ln 237-239, how come these three variables were used in the MLR? Comes currently a bit out of the blue and needs to be better motivated.

*Hydrology, terrain roughness, and vegetation productivity were included as covariates because they are known to be among the primary landscape controls on DIC cycling. We have clarified this and included citations.*

Ln 239-245, again it is hard to judge the certainty in this modelling effort given the already above raised concern about the Q estimation.

*Please see our reply to your comment for Ln 153-160. We acknowledge there is some uncertainty. Yet, this approach enables us to generate a first estimate of the relevance of RTSs in*

*carbonate alkalinity production and export relative to other landscape conditions known to influence DIC in fluvial networks.*

Ln 259-, I guess very much a question of personal taste but I feel the ms do not benefit from the mixing of results and discussion. It would be easier to keep focus by separating them in my opinion.

*We agree with you and Reviewer #1 about this. We restructured the manuscript so that the Results and Discussion are presented separately.*

Ln 278, I am not familiar with the given reference, but what is meant by "regional carbonate"? Also in this couple of sentences, I agree with the overall argumentation, but can you completely rule out a biotic source contribution? The fractionation between carbonate and CO2 (8‰ is rather theoretical. Could a mixing with geogenic and bio- genic IC be possible for generating 13C-CO2 of -11.4 to 12.1‰ You have a substantial DOC pool which is also cited by being "relatively biolabile".

*This was meant to read "… regional carbonate bedrock". Revised. Yes, good point about $CO_2$ being a mix of biogenic and geogenic sources. We have added brief text clarifying that these isotopic values may also reflect some contribution of $^{13}C$-depleted $CO_2$ from biogenic sources.*

Ln 285, how CH4 was sampled is mentioned in the methods but from what I see this is the only place where any data is presented, and then very shortly. Maybe the data is saved for another story but I believe it would further strengthen the story if it could be included for example in table 1 and with subsequent incorporation in the text.

*We tell the $CH_4$ story in an earlier publication (Zolkos et al. 2019, e.g. Sections 3.1, 3.2, 3.3, 4.1, 4.3, 4.4). $CH_4$ measurements in this study were done to assess for potential effects from methanogenesis on stable $CO_2$ isotopes.*

*Zolkos, S., Tank, S. E., Striegl, R. G., & Kokelj, S. V. (2019). Thermokarst Effects on Carbon Dioxide and Methane Fluxes in Streams on the Peel Plateau (NWT, Canada). Journal of Geophysical Research: Biogeosciences, 124(7), 1781-1798.*

Ln 310-313, yes it could be due to adsorption to RTS sediments, but I guess it could also be due to lower mineralization than degassing rates. Might be worth to mention.

*Yes, good point. We have clarified this in the text.*

Ln 347-349, is it really clear that biotic CO2 were the primary source of DIC in the headwaters of Stony Creek? Could not geogenic sources still be highly influential? The 13C-DIC and 13C-CO2 values (-11.6 and -13.8‰ respectively) points towards a biogenic/geogenic mixing, or?

*Fair question. As noted, mixing between the stream and atmosphere was a primary $CO_2$ source. This is supported by the values from other measurements, such as $pCO_2$ at an approximately atmospheric level and also low $HCO_3^-$ and pH. The former suggests relatively minor biotic $CO_2$ production (organic matter mineralization) and/or greater effects from degassing on $CO_2$ than from biotic processes. The latter suggests stronger effects on DIC speciation from variability in pH ($CO_2 > HCO_3^-$) than from mineral weathering. Together, these results suggest that geogenic mixing is a less parsimonious explanation.*

Ln 403-405, do the study really evaluate "across gradients of thermokarst disturbance"? I believe something like influence of RTS on IC cycling and how this signal is propagated across different fluvial scales is better describing the story.

*Good point. Revised.*

Ln 419-434, I somehow miss the full interpretation of the findings of the current study for the large scale picture. How do you suggest your results should be considered in large scale estimates, i.e. how does it affect the previous judgement of the area as a "modest source of CO2".

*Fair point. We consider the broader relevance of our findings in our revised Discussion and we provide an updated conceptual model to help generalize our findings.*

A general question: how common are RTS across permafrost regions worldwide? How applicable are the findings here for other areas?

*Good question. We consider this in the Introduction and we believe that our edits help to clarify this.*

Figure 1. For a non-north American reader, a more large-scale inset of where the area is found would be appreciated.

*Good suggestion, thanks. Figure revised.*

---

## Author Comment (AC3) · 13 Jul 2020

**Anonymous Reviewer #1**

Zolkos et al., present a high quality characterisation of running water chemistry in a sub-catchment of the Peel River where the aim was to determine the effect of retrogressive thaw slump (RTS) on DIC sources and export. The study design, incorporating three transects at different spatial scales (1. retrogressive thaw slump (RTS) runoff water, 2. an intermediate size catchment with direct fed with RTS runoff, 3. A large catchment fed indirectly with RTS runoff through its tributaries), is an interesting and innovative sampling approach. The dataset, including a large number of key variables, is also of very high quality. The research question is also highly relevant to our understanding of the permafrost climate-feedback. The author rightfully stated that changes in carbonate alkalinity export in response to permafrost degradation has been far less studied that those for organic carbon and carbon dioxide. The study is also taking place in a relatively understudied region, which makes it even more valuable. While the subject and design of the study is of high quality, I find that the discussion and conceptualisation of the result need significant improvement. In short, the paper does not make full use of its potential.

The novelty of the paper lies in the approach of scales on the effect of RTSs on DIC export. At the moment, these three scales are taken separately, into three almost individual studies. Is the message simply that we can perceive the RTS effect at each of those three scales or is there a greater interpretation of how these effects integrate with increasing spatial scale and decreasing land-water connectivity? The paper would have more impact if the author could conceptualise these results and formulate how RTSs affect DIC cycling across scales rather than simply testifying that it has an influence. For example, does the "RTS effect" amplifies, is conserved, accumulates or becomes diluted with increasing scale. The author already documented that RTSs alter riverine DIC cycling in a previous publication in GRL (Zolkos et al, 2018). At present, this manuscript adds little to this state of knowledge, but this could be remediated by conceptualising further the effect of scales.

I have made a few suggestions to improve the presentation of results and conceptualisation of the discussion.

*Thank you for the thoughtful and constructive feedback. Regarding your questions and comments above: Good question about the nature of RTS effects across scales (e.g. is it amplified, conserved, diluted, etc.). Effects on DIC are clearly amplified in headwaters. While $CO_2$ diminishes rapidly within headwaters, the $HCO_3^-$ signal is preserved at broader scales. We use circumstantial evidence to infer the latter point in Zolkos et al. (2018), which focuses on biogeochemical effects immediately downstream of RTSs and leverages long-term Peel River data to assess changes over time. Our current study investigates processes occurring between headwater streams and the larger Peel River. We therefore contend that our current manuscript builds on our 2018 GRL paper and improves understanding of RTS effects on C cycling. We believe our edits help to clarify these points, in part by addressing your helpful suggestion to better conceptualize the effects of scale. Below, please find replies to your comments.*

The sampling design is interesting and valuable, but arguably difficult to communicate to the reader. Having the Results and Discussion section together makes it even harder for the reader to put together the key results, and follow the discussion points that are mixed through the text. I recommend to separate these two parts. The first part of the results and discussion section details the changes in water chemistry in each of the three transect. The first part of the Result and Discussion section (section 3.1, 3.2 ad 3.3) details the water chemistry patterns in each transect with discussion points mixed through the text. Having this structure increases the impression that this study actually involves three separate studies rather than one. I suggest to also structure the results by water chemistry variable rather than sites. The study measured a large number of important and interesting water chemistry variables. Each should be presented clearly in the result section for the reader to identify. Each sections should provide, among other things, the overall range in values for the whole study, compare this range between each of the three scales and within each transect.

*Great points. Thank you for the helpful suggestion. We restructured the manuscript so that Results and Discussion are presented separately. In the revised Results section we discuss results by hydrochemical parameter and present ranges of values, as you suggest.*

Rather than naming the sites by their official river name, why not call them with a more conceptual name that represents the idea behind the sampling design. I like that the symbols in figures have numbers to indicate their position along the transect, but RTS FM2 site DC and SC has little meaning for the reader. I have also provided a few suggestions below to format the figures in a more visually telling way.

*We appreciate the suggestion, but will use the official names for continuity with the literature. Our edits were made with this in mind, to improve demarcation between the different watershed scales.*

Playing with your dataset I attempted to trace the d13CO2 source with the miller-trans plots, I found a clear difference in that value for the RTS runoff site (d13Csource -12‰ and the DC and SC rivers (-22.9‰. This suggest two predominant CO2 sources in this catchment and those end-members could potentially be used for calculations. The RTS site is consistent with a geogenic CO2 source, while the rivers have a predominant biogenic soil CO2 source. Would it be worth including this kind of approach to your results?

*Thanks for the suggestion. We explored Miller-Tans plots, but do not have enough samples from undisturbed headwaters (n = 3) to trace $CO_2$ sources with this approach. We agree that $\delta^{13}C$-$CO_2$ values reflect at least two predominant $CO_2$ sources in the catchment. Together, our $\delta^{13}C$-$CO_2$ values and $\delta^{13}C$-DIC vs. pH plot reflect spatial trends in DIC and $CO_2$ sources (atmospheric, soil biotic, mineral weathering). Building on this plot and following your suggested changes to the conceptual diagram, we clarify the relative importance of biotic versus mineral weathering sources for $CO_2$ and DIC, and how sources may change with movement downstream.*

The effect of scales, with a varing degree of terrestrial connectivity, is only discussed in the context of circumpolar region with, but I believe that the study should also be put in the context of the broader literature, including lower latitude catchments, where many studies have also examined the effect of changing land-water connectivity with size.

*Yes, good point. We added brief text towards the end of the manuscript which considers this.*

I find it interesting to see the downstream changes in HCO3 concentration. It could be worth mentioning that studies modelling stream CO2 evasion based on d13CO2 value assume that carbonate alkalinity is conserve in river networks (Polsenaere 2012 Geochimica et Cosmochimica Acta and Venkiteswaran 2014 PLOS one).

*Good point. Though, this assumption will not hold true where changes in catchment lithology, groundwater inputs, etc. influence carbonate alkalinity. Nevertheless, it is interesting, so we have added brief text considering this.*

The term thermokarst and retrogressive thaw slumps are used interchangeably, I suggest to stick to retrogressive thaw slumps since this was the focus of the study and the findings may not be applicable to all kinds of thermokarst disturbances.

*Thank you for the suggestion. We have replaced thermokarst with retrogressive thaw slump where we discuss trends specific to the Peel Plateau. We retain "thermokarst" when we are discussing these effects from a broader perspective.*

The term "abiotic-inorganic processes" is a vague term to me, what are the processes included in that? Carbonate equilibrium reaction and CO2 evasion? Could you be more specific? Either a more specific term should be used or the choice of terminology should be justified in the text.

*Clarified.*

Abstract: Line 21:26: The results/discussion section of the abstract list changes in water chemistry in each of the three transect. I believe this section, and other relevant section in the main text, should come up with a more unified message of the RTS effect across scales, rather than at individual scales.

*Thank you for the suggestion. Please see our reply to your major comment, above.*

Line 30: I have trouble with the word "prevalent" here since it implies that one is larger than the other, while such calculations have not been done in the study.

*Revised.*

Introduction: Lines 32-33: This first sentence would introduce better the second paragraph where the source/sink relationship of DIC is detailed. This first paragraph discusses how the arctic landscape is changing and what is known of its influence on DIC export.

*Thanks for the helpful suggestion. Revised.*

Lines: 41-44: Since we include CO2 in this pool, the increase in soil respiration and/or increase in aquatic DOC degradation should also be part of this list. The citations be separated to indicate which if the listed mechanisms have been highlighted by each study.

*Revised.*

Lines 47-64: This is mostly textbook material and could be synthesised. I think what you want to express is why it's important not only to account for the mass of DIC exported but also to identify its sources DIC. Without this we can't determine whether this is a new sink or source of C in the short and long term perspective. I would suggest to move some of this information in section 2.5 of the methods and rework this paragraph to emphasise the simply the importance of source separation for DIC.

*Fair point. We have modified the text but choose to discuss much of the original concepts, because our manuscript intends to target permafrost biogeochemists, who typically consider processes over contemporary timescales and processes associated with organic matter decomposition. So, this "textbook" information is provided – in part – to emphasize to this community the importance of incorporating weathering processes into consideration of the short- and long-term effects of permafrost thaw on biogeochemical (especially carbon) cycling.*

Line 65: Could you state the representativeness of this study, which areas across the circumpolar north could be similar to your studied location?

*Revised.*

Line 94: This is a very important point which I think should be stated earlier in the introduction and given more emphasis.

*Text added earlier in the Introduction to elaborate on this point.*

Methods Line 133: How deep are those rivers? Can stream order be provided somewhere too.

*We were not equipped with instruments to safely measure the depth of the rivers and unfortunately this information is not known. We have added stream order to Appendix Table A1.*

Section 2.3 This is a nice way to work around flow measurement scarcity in this region. But should this section come after section 2.6?

*Yes, this is more closely related to statistics and logically could be placed just before Sec. 2.7. Revised.*

Line 212: What threshold was used in the flow accumulation to delineate the stream lines and catchment boundary? Was it validated with the areal photos mentioned on line 155 or something else?

*We have clarified these methods in our 'Geospatial Analyses' section. Briefly, geospatial data were validated by modifying as needed to align with stream networks visible in moderate-resolution (10 m) Sentinel-2 satellite imagery, following St. Pierre et al. (2018).*

*St. Pierre, K. A., Zolkos, S., Shakil, S., Tank, S. E., St. Louis, V. L., & Kokelj, S. V. (2018). Unprecedented increases in total and methyl mercury concentrations downstream of retrogressive thaw slumps in the western Canadian Arctic. Environmental Science & Technology, 52(24), 14099-14109.*

Section 2.4, shouldn't this section be called water chemistry analysis?

*Changed to "Hydrochemical Analyses".*

Line 245: Why not model the full carbonate alkalinity pool (HCO3+ CO3)? Arguably the CO3 pool is small at this pH range and shouldn't make a much difference to the model, but I find this conceptually more appropriate.

*We agree that using $[HCO_3^- + CO_3^{2-}]$ is conceptually more appropriate. We re-ran the model using carbonate alkalinity and yes, it is essentially identical to using only $HCO_3^-$. Nevertheless, we revised the results (Table 2) and text to reflect the updated model.*

Results and Discussion

Section 3.1, 3.2 and 3.4 starts with a sentence stating how the author interprets DIC sources and cycling in each transect. This seems odd to me. I would rather the author takes me to that conclusion by presenting the results first.

*Thank you for the suggestion. We believe that our revisions, which include splitting the Results and Discussion, address your suggestion.*

Line 275: The Miller-tans analysis supports that as well. Again, this statement comes before presenting the key results.

*Although we have not done a Miller-Tans analysis, for the reasons described above, our edits were done with an eye to elucidating the processes and sources which contribute to DIC in this system.*

Line 290: Is the term "geogenic" more appropriate?

*Revised.*

Line 285: Probably right, but do you have any measurements or estimates of CH4 concentration at the source - in the groundwater? The CH4 might have evaded already, but its imprint on d13CO2 values might still be there.

*Unfortunately, we do not have any measurements of $[CH_4]$ in groundwater.*

Line 301: That is also supported with the Miller-tans. But this biogenic soil CO2 source seems to prevail in other sites as well albeit with some mixing with the geogenic source. Could mixing between the H2SO4 weathering and biogenic soil CO2 be back calculated?

*$n = 1$ in the Dempster Creek headwaters, so it is supported by the Miller-Tans analysis when data from all sampling points are considered. Also, $[CO_2]$ and stable isotopes alone would provide only a very rough estimate of the proportions of biogenic $CO_2$ and abiotic $CO_2$, the latter of which may also be sourced from $H_2CO_3$ carbonate weathering ~ DIC speciation reactions, and atmospheric $CO_2$. As we lack the measurements (e.g. $^{14}C$ of $CO_2$) to make more robust estimates of $CO_2$ contributions from these varied sources, we do not attempt this.*

Line 314: This is also demonstrated in headwater streams at other latitudes and should be mentioned as well.

*Yes, good point. Elaborated and citations added, such as:*

*Campeau, A., Lapierre, J. F., Vachon, D., & del Giorgio, P. A. (2014). Regional contribution of CO2 and CH4 fluxes from the fluvial network in a lowland boreal landscape of Québec. Global Biogeochemical Cycles, 28(1), 57-69.*

*Hutchins, R. H., Prairie, Y. T., & del Giorgio, P. A. (2019). Large-Scale Landscape Drivers of CO2, CH4, DOC, and DIC in Boreal River Networks. Global Biogeochemical Cycles, 33(2), 125-142.*

Line 314; Again I find this term "abiotic inorganic" to be vague. And what do you mean by dominates? Abiotic dominates biotic, or inorganic dominates organic?

*Clarified.*

Line 351: What does "amplified inorganic carbon cycling" means?"

*Clarified.*

Line 371: The term biotic is used here, but could the DOC be degraded photochemically as well?

*Although photochemical degradation of DOC could lower $\delta^{13}C$-$CO_2$ values (references below), these streams are very turbid and therefore photodegradation is likely limited.*

*Opsahl, S. P., & Zepp, R. G. (2001). Photochemically‑induced alteration of stable carbon isotope ratios ($\delta13C$) in terrigenous dissolved organic carbon. Geophysical Research Letters, 28(12), 2417-2420.*

*Vähätalo, A. V., & Wetzel, R. G. (2008). Long‑term photochemical and microbial decomposition of wetland‑derived dissolved organic matter with alteration of 13C: 12C mass ratio. Limnology and Oceanography, 53(4), 1387-1392.*

Line 380: Is the model intended only to bring evidence to the fact that RTS increase alkalinity export or for a possibly larger modelling/budget exercises? Could you make use of that model already in this paper for a final "wrap up" exercise?

*The model was intended to assess the influence of various landscape characteristics (including RTSs) on carbonate alkalinity yields (please see Section 2.7). We refrain from exercises on larger budgets of carbonate alkalinity, as our model reflects a snapshot of DIC dynamics within a relatively local area.*

Line 393: Dominate over what?

*Revised.*

Line 405: This "conceptual model of land-freshwater linkages" needs more elaboration. As it stands, this model seems more like a list of DIC sources and sinks across this catchment than a generalisable model. A starting point would be to determine how does it integrates with other models at lower latitudes? How do changes in DIC sources and sinks caused by RTS integrates with other water chemistry properties and C species (organic vs inorganic) that were documented in other studies?

*Good point and, as noted in our reply to your comment on Figure 8 (below), we agree that the current conceptual diagram is fairly specific to the Peel Plateau. We believe the revised diagram helps to generalize our findings and, with revisions to the text, address your comments here.*

Line 410: What exactly was "striking"?

*Clarified.*

Line 414-418: How does this conclusion differ from the authors previous publication cited here?

*A key difference is the nested watershed/transect approach, which allows us to elucidate mechanisms and how they change downstream. Clarified.*

Line 432: Should your model be used to that effect? If so, it should be stated.

*Revised.*

Line 443: " change for C cycle in the rapidly changing arctic landscape"

*Revised.*

Authors Contribution What did D.O. do?

*All authors contributed to manuscript writing. Clarified.*

Figures and Tables

Table2: Should these values be presented in supplementary and only the model be presented in this table? I find the second part of the table easy to miss.

*We feel that the values in the upper panel are important to include in the main text and agree that the lower panel may be easy to miss. To address this, we revised the first sentence of the table caption to draw attention to both the upper and lower panels.*

Figure 1: The map feels quite dense, can the photos and context map be placed outside? Could the bedrock lithologies be illustrated on the map?

*Thank you for the helpful suggestion. We revised the map and the individual components now have more space. Unfortunately, there are no high-resolution GIS data of bedrock lithology available.*

Figure 2: The figures should be placed vertically rather than horizontally since they have the same x-axis. Also in caption, please clearly state that the top x-axis is for the dempster creek transect while the second is for the stony creek. Could the points for each transect be connected with a line for visualisation. Could HCO3 and CO2 concentration be on the same unit? Could there be a third 3 axis for the distance along the RTS runoff transect?

*Good suggestion to consolidate x-axes by stacking the figures. Revised. Following your other helpful suggestion, we have also included DOC and $SUVA_{254}$ in Fig. 2.*

*Caption revised to distinguish between the RTS FM2 runoff, Dempster Creek, and Stony Creek x-axes.*

*We choose to omit lines connecting the points and, as noted in the caption, instead label consecutive downstream points with their corresponding site numbers; we reserve lines to indicate significant trends.*

*We like and appreciate your suggestion to report $CO_2$ in µM, which could help to facilitate a more direct comparison with the $HCO_3^-$ concentrations shown in Fig. 1a. However, for Fig. 1b, we choose to report $CO_2$ in units of partial pressure rather than µM, to make a more intuitive link with atmospheric $CO_2$. For the readers' reference, $CO_2$ in µM is reported in Table 1.*

*We have added a third axes for RTS FM2 runoff and plotted the data (as in Table 1).*

Figure 4: Can you give a reference to these end-members

*Thank you for the suggestion. In the caption, we added a reference to Methods Sec. 2.4, in which we briefly elaborate on these methods and cite a publication used to derive the end-members (Zhang et al. 1995).*

*Zhang, J., Quay, P. D., & Wilbur, D. O. (1995). Carbon isotope fractionation during gas-water exchange and dissolution of $CO_2$. Geochimica et Cosmochimica Acta, 59(1), 107-114.*

Figure 5: Open vs closed symbols would be clearer perhaps?

*Thank you for the suggestion. This does appear to help. Figure revised.*

Figure 6 and 7: Should these two figures be merged with Figure 2? This would help draw a more complete picture of simultaneous changes in water chemistry along the transects. Why isn't there a similar figure for d13C-DIC values?

*Good idea. We have included the DOC and SUVA$_{254}$ plots as part of Fig. 2. We choose not to replicate this plot style for $\delta^{13}$C-DIC, because the downstream trends in $\delta^{13}$C-DIC are shown in Figures 4 and 5.*

Figure 8: This is a nice schematic, but it limits the scope of the study. The schematic mostly lists the sources and transformation of DIC in this catchment. Does it only applies to this catchment, i.e. was the goal to map those processes, or can it be generalised to other catchments? I think this figure could be useful if it was to conceptualise the effect of RTS across scales, not just make a summary of all the processes identified in the data for this specific catchment. I have in mind something along the lines of Hotchkiss et al. 2015 NatGeo Figure 3.

*Thank you, we agree that the figure is somewhat limiting and we appreciate your suggestion to broaden the scope of our conceptual diagram. We believe the revised conceptual diagram helps to generalize our findings.*

Table A1. Why not keep the distance units the same and just add decimals for FM2 site.

*Yes, good point! Revised.*

Table A2: I find this could be useful in the main manuscript since it also provides a list of the DIC sources you are trying to separate.

*Good suggestion. Added to main text.*

---

## Author Response (AR2)

**Thermokarst amplifies fluvial inorganic carbon cycling and export across watershed scales on the Peel Plateau, Canada**

Scott Zolkos, Suzanne E. Tank, Robert G. Striegl, Steven V. Kokelj, Justin Kokoszka, Cristian Estop-Aragonés, David Olefeldt

**Point-by-point response to reviewers**

Dear Dr. Park,

Wonderful news! Thank you for your feedback on our revised manuscript. As detailed in our replies below, we have addressed the requested technical corrections. Please find our replies in blue text.

Sincerely,
Scott Zolkos, on behalf of the co-authors

**Comments to the Author**

Dear Authors,

Thank you for your careful revision of the manuscript considering all reviewer comments and suggestions. I am pleased to let you know that your manuscript can be published after technical corrections of the following:

Please provide a brief explanation or definition of RTSs at the first use of the term in Abstracts and Introduction.

Thank you for the helpful comment. We have modified the text in the abstract to more clearly define RTSs. It now reads: "…we determine the effects of slope thermokarst in the form of retrogressive thaw slump (RTS) activity…". (please refer to L20–21)

In the Introduction, we state that changes in the Arctic are occurring "… via hillslope thermokarst features, the largest of which include retrogressive thaw slumps (RTSs)." We contend that the acronym "RTS" and what an RTS is are both defined here, as we describe RTSs as "hillslope thermokarst features". In support of this explanation, we define thermokarst earlier in the Introduction (please refer to L53) as "… terrain consolidation following thaw (thermokarst)…".

Line 171: The original name of the program is "CO2SYS".

Thank you for noticing this. Text revised.

Lines 369-370 "In the Stony Creek headwaters, we also found an influence from exchange with atmospheric CO": indicated by which result?

Thank you for the helpful comment. We have clarified the text to indicate the results which support this finding. The text now reads: "In the Stony Creek headwaters, intermediate $\delta^{13}$C-$CO_2$ and $p$CO$_2$ saturation suggested influence from exchange with atmospheric $CO_2$ and from some biogenic $CO_2$."

Tank et al. (2020): Is this a paper published in the journal "Permafrost and Periglacial Processes"? Please correct the journal name and provide information about the volume and pages in a proper format.

Thank you for bringing this to our attention. Revised.

Sincerely,

Ji-Hyung Park
Associate Editor, Biogeosciences

[revised manuscript text omitted]